# Entorhinal-retrosplenial circuits for allocentric-egocentric transformation of boundary coding

Joeri BG van Wijngaarden[1], Susanne S Babl[2], Hiroshi T Ito[1]*

[1]Max Planck Institute for Brain Research, Frankfurt, Germany; [2]Institute of Neurophysiology, Neuroscience Center, Goethe University, Frankfurt, Germany

**Abstract** Spatial navigation requires landmark coding from two perspectives, relying on viewpoint-invariant and self-referenced representations. The brain encodes information within each reference frame but their interactions and functional dependency remains unclear. Here we investigate the relationship between neurons in the rat's retrosplenial cortex (RSC) and entorhinal cortex (MEC) that increase firing near boundaries of space. Border cells in RSC specifically encode walls, but not objects, and are sensitive to the animal's direction to nearby borders. These egocentric representations are generated independent of visual or whisker sensation but are affected by inputs from MEC that contains allocentric spatial cells. Pharmaco- and optogenetic inhibition of MEC led to a disruption of border coding in RSC, but not vice versa, indicating allocentric-to-egocentric transformation. Finally, RSC border cells fire prospective to the animal's next motion, unlike those in MEC, revealing the MEC-RSC pathway as an extended border coding circuit that implements coordinate transformation to guide navigation behavior.

## Introduction

Animals use landmarks in the environment as references to identify the self's position and a destination in space. Rodents, for example, are able to discriminate positions within an open field arena by relying on distal cues in the room, allowing them to navigate to a desired location (*Morris, 1981*). This ability is manifested in the activity of neurons that fire at particular locations in space, such as place cells or grid cells (*Hafting et al., 2005*; *O'Keefe and Dostrovsky, 1971*), and population activity of place cells can distinguish nearby positions at several centimeter resolution in an open field arena (*Brown et al., 1998*). It has been suggested that this accurate spatial coding is based on the estimation of distance and direction relative to landmarks, particularly environmental boundaries such as walls or edges (*Barry et al., 2006*; *O'Keefe and Burgess, 1996*). For example, a subpopulation of neurons in the medial entorhinal cortex (MEC) or the subiculum increase firing rates near the environmental boundaries, called border cells or boundary-vector cells (*Lever et al., 2009*; *Solstad et al., 2008*). The presence of dedicated representations of environmental borders in the hippocampus and parahippocampal regions implies a pivotal role of boundary information in generating accurate spatial representations in the brain. In accordance with this idea, border cells in MEC develop earlier than grid cells after birth, exhibiting adult-like firing fields at postnatal days 16–18 when grid cells still exhibit immature irregular firing fields (*Bjerknes et al., 2014*). It has further been shown that position errors of firing fields of grid cells accumulate after the animal leaves a wall of an open field arena, suggesting an error-correcting role of environmental boundaries for internal spatial representations (*Hardcastle et al., 2015*).

While neurons in MEC or subiculum represent environmental boundaries in a viewpoint-invariant allocentric coordinate frame, recent studies have identified neurons that encode various spatial landmarks in a self-centered egocentric perspective, for example, objects in the lateral entorhinal cortex

*For correspondence:
hiroshi.ito@brain.mpg.de

Competing interests: The authors declare that no competing interests exist.

(*Wang et al., 2018*), the arena center in the postrhinal cortex (*LaChance et al., 2019*) and nearby boundaries in the dorsomedial striatum (*Hinman et al., 2019*), postrhinal cortex (*Gofman et al., 2019*), and retrosplenial cortex (*Alexander et al., 2020*). These neurons exhibit spatial tuning relative to the self's body, providing information about the direction and distance of landmarks from the animal's viewpoint. The brain thus forms landmark representations in two different reference frames, in either egocentric or allocentric spatial coordinate systems. It is however still largely unclear how each type of representation is generated, and the degree of functional interaction and dependency that exists between them.

Self-referenced representations can be generated directly by sensory inputs, as incoming sensory information is initially anchored to sensory organs mapped on the body, for example, visual information on the retina, tactile sensation on the skin, or proprioceptive signals from skeletal muscles. This information may then be used to constitute viewpoint-invariant allocentric spatial representations, or a cognitive map (*Hafting et al., 2005*; *McNaughton et al., 2006*; *O'Keefe and Nadel, 1978*), implying the importance of transformation from egocentric to allocentric coordinate frames. It is however also possible that egocentric representations are the result of a coordinate transformation of the brain's allocentric map to guide the animal's behavior, because navigational plans and their underlying sequence of motor actions based on the internal map should be executed in the brain's motor areas in self-referenced coordinates (*Ekstrom et al., 2014*; *Georgopoulos, 1988*). While previous studies suggested the importance of coordinate transformation for spatial navigation relying on the brain's allocentric map (*Bicanski and Burgess, 2018*; *Byrne et al., 2007*; *Clark et al., 2018*), the direct evidence of such transformation in the brain is still missing. Furthermore, clarification of the direction of transformation, either from allocentric to egocentric or vice versa, is necessary to understand the roles of egocentric landmark coding in navigation.

In the present work, we address the functional relationships of boundary representations in different coordinate frames that exist in reciprocally-connected brain regions, the retrosplenial (RSC) and medial entorhinal cortex (MEC). We first established a quantitative metric to characterize a subpopulation of neurons in RSC that increase their firing rates near environmental borders. Unlike border cells in MEC, those in RSC fire indiscriminately to all walls, and a subset of them are additionally modulated by the animal's head-direction relative to the closest wall, providing information about the distance and direction to nearby boundaries in an egocentric coordinate frame. We explored under which environmental circumstances this information is generated to determine the impact of sensory and spatial cues on boundary coding in RSC. We then examined the functional dependence between border cells in MEC and RSC with pharmacogenetic and optogenetic inactivation techniques, clarifying coordinate transformation between the regions. Lastly, by applying decoding and spike information metrics, we demonstrate that firing of border cells in RSC exhibits additional correlates with the animal's next movements, revealing the role of the MEC-RSC pathway in interfacing the brain's allocentric map with navigation behaviors.

## Results

### RSC cells fire near the maze perimeter at specific distances

We performed electrophysiological recordings of neuronal activity in RSC (*Figure 1A*, *Figure 1—figure supplement 1*) of rats as they explored a squared open field arena and foraged for scattered chocolate pellets (*Figure 1B*). All animals were sufficiently habituated to the environment and procedures, and actively explored the entire arena (*Figure 1C*). The experimental setup was placed in the room with fixed landmarks to allow the animals to orient themselves relative to external features.

We recorded the activity of 5415 RSC neurons across eight animals (n = 82 sessions) and observed a subpopulation of cells that fired consistently at the edge of the arena (*Figure 1C*). Across this subgroup, there was a variety of preferred firing distances from the wall, ranging from very near proximity up to a body-length (15–18 cm) away. Unlike traditional border cells found in MEC and subiculum (*Solstad et al., 2008*; *Stewart et al., 2014*), these border responses occurred throughout the environment on each of the four available walls. RSC border cells furthermore form multiple firing fields that are not necessarily directly connected to the wall. Typical border cell classification using the original border score (*Solstad et al., 2008*) identified only a small fraction of border cells in RSC, as this score is based on the occupancy of a single firing field along a wall and is strongly

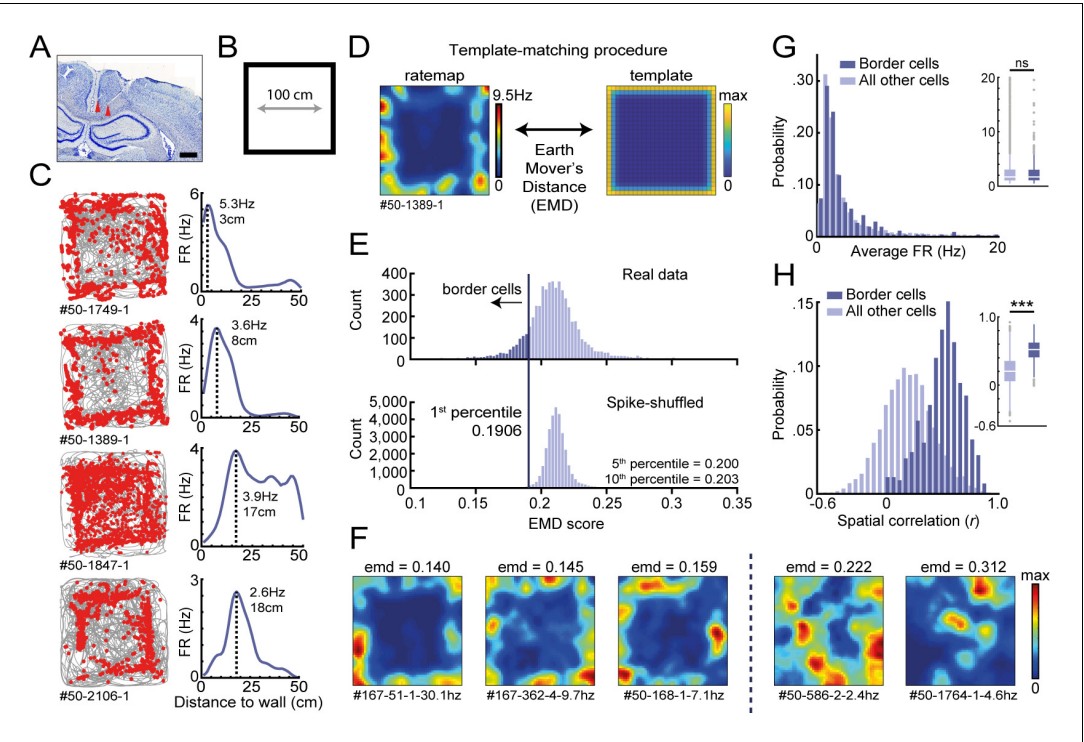

**Figure 1.** Response profiles of border cells in RSC. (**A**) Location of tetrode tracts marked with red in an example Nissl-stained coronal section. Scale bar, 1000 µm. (**B**) Task behavior consisted of free exploration in a squared 1 m² arena. (**C**) Trajectory spike plots (left column) and distance firing rate (FR) plots (right column) of four example cells that fired at different distances away from the wall, relative to the closest wall at any time. Gray lines indicate the animal's trajectory and red dots the rat's position when a spike occurred. (**D**) A template-matching procedure was applied to classify border cells by calculating the Earth Mover's Distance (EMD) between each cell's spatial rate map and an ideal template (see Materials and methods). (**E**) A cell was classified as a border cell when its EMD score was below the 1st percentile of a shuffled null distribution, together with an average FR above 0.5 Hz. (**F**) Color-coded spatial rate maps of five example cells with different EMD scores, where warm colors indicate high firing. From left to right: three typical border cells, a non-uniform firing cell, and a cell with focused firing fields. (**G**) Distribution of average FR over the entire recording day shows no difference between border cells and other recorded cells. (**H**) Distribution of spatial correlations between recorded sessions shows significantly higher spatial correlations for border cells compared to other recorded cells. ***p<0.001, Wilcoxon ranksum test.

The online version of this article includes the following figure supplement(s) for figure 1:

**Figure supplement 1.** Nissl-stained coronal sections showing recording locations and tetrode tracts for all recording experiments.

**Figure supplement 2.** Comparison between the EMD metric and original border score and its relationship with a cell's firing rate.

**Figure supplement 3.** The dissociation between the animal's running speed and activity around borders.

**Figure supplement 4.** RSC border cell firing under novel conditions and different arena shapes.

---

biased to connected bins (*Figure 1—figure supplement 2A–C*). We thus developed a new model-based approach using a template-matching procedure to classify these border cells in RSC (*Figure 1D–F*), based on *Grossberger et al., 2018*.

This method uses two-dimensional (2D) information of the firing rate maps and builds on the assumption that border cells have their spikes concentrated at the entire outer ring of the arena, incorporating geometric information into the classification procedure. The dissimilarity between a cell's spatial firing rate map and a 'boundary' template (*Figure 1D,E*) was assessed by the algorithm based on the Earth Mover's Distance (*Hitchcock, 1941*; *Rubner et al., 1998*) (EMD; see

Materials and methods), a distance metric from the mathematical theory of optimal transport. The EMD is a normalized score, calculated as the minimum cost required to match a cell's firing rate map with the template distribution by moving firing rate units, also known as the Wasserstein distance. While this metric is sensitive to a change in the geometric shape of rate maps, it is robust to small variations of preferred firing distances or pixel-by-pixel jittering, giving a single metric that can quantitatively assess changes in the cell's spatial tuning as a function of experimental manipulations, such as reshaping of the maze or adding an object or wall.

Border cells were defined as cells with a low dissimilarity EMD score below the 1st percentile of a spike-shuffled null distribution of 0.191, and an average firing rate (FR) above 0.5 Hz. Null distributions of EMD scores were invariant to the overall FR of cells included, giving a consistent criterion across cells (*Figure 1—figure supplement 2F*). In total, 485 out of 5415 RSC cells (9.0%) passed this criterion (*Figure 1E,F*, *Figure 1—figure supplement 2G*), where neurons showed no relationship between their overall spiking rate and associated EMD score (Pearson's correlation: r = 0.037, p=0.42; *Figure 1—figure supplement 2D*). Selected border cells had a similar distribution of average firing rates compared to other recorded cells (border cells: FR = 2.97 ± 0.20 Hz, others: FR = 3.25 ± 0.07 Hz; Wilcoxon ranksum test: z = 0.057, p=0.955; *Figure 1G*), but had significantly higher spatial correlations between the first and last recording sessions (border cells: r = 0.52 ± 0.008, others: r = 0.20 ± 0.003; Wilcoxon ranksum test: z = −24.50, p=1.60 × 10$^{-132}$; *Figure 1H*). Border cells were recorded across the granular and dysgranular regions of the RSC, and had waveform properties similar to other cells recorded on the same tetrodes (*Figure 1—figure supplement 3A,B,D,E*). Border cells showed activity already from the beginning of the session and did not need time for adaptation in a novel arena and experimental room (first-half familiar session, FR = 1.98 ± 0.42 Hz; first-half novel session, FR = 2.30 ± 0.53 Hz, p=0.56; second half novel session, FR = 1.71 ± 0.54 Hz, p=0.68; Wilcoxon signed-rank test; n = 14 border cells; *Figure 1—figure supplement 4D–G*). While firing of RSC border cells was additionally modulated by the running speed of the animal, with lower firing rates in the low-speed range (p<0.05 in the range of 0–12 cm/s; Wilcoxon ranksum test; *Figure 1—figure supplement 3G*), this modulation could not account for the cell's spatial tuning, as the animal's average speed was uniform across space and unrelated to the distance to any boundary (non-significant in the range of 5–45 cm; Wilcoxon ranksum test against overall median; Bonferroni-corrected α = 0.005; *Figure 1—figure supplement 3F,H*). Applying an unbiased classification approach based on linear-nonlinear models (*Hardcastle et al., 2017*) showed that running speed and wall-distance are two independent factors that explain the activity of RSC cells, confirming that RSC border and speed tuning are separately expressed features (*Figure 1—figure supplement 3C*).

## Border cells form new firing fields nearby added walls but not objects

To understand the generation of boundary coding in RSC, we quantified the impact of the change of environmental features on the activity of RSC border cells by using the EMD metric. We first asked if the firing of these border cells is limited to walls, or whether these cells also encode information about other features of the environment, such as local cues or objects (*Høydal et al., 2019*; *Jacob et al., 2017*). Our first manipulation was to temporarily add a wall, protruding from one side into the center of the maze (*Figure 2A,B*). Border cells formed new firing fields around the added walls accordingly, as their firing rate inside a region-of-interest (ROI; 15 × 5 spatial bins) around the wall increased significantly in the added wall sessions (Regular: FR = 1.19 ± 0.13 Hz; Added wall: FR = 1.58 ± 0.21 Hz; Wilcoxon signed-rank test: z = −2.67, p=0.0076; n = 42 border cells; *Figure 2C*). This was accompanied by a sharp drop in spatial correlations between rate maps of regular versus added wall sessions (Reg-Reg: r = 0.51 ± 0.04, Reg-Wall: r = 0.25 ± 0.04; Wilcoxon signed-rank test: z = 4.43, p=9.31 × 10$^{-6}$; Bonferroni-corrected α = 0.025; *Figure 2D*), while correlations remained high when comparing within session types (Wall-Wall: r = 0.47 ± 0.03; Wilcoxon signed-rank test with Reg-Reg correlation: z = 0.63, p=0.53; Bonferroni-corrected α = 0.025; *Figure 2D*). The EMD metric furthermore showed a significant increase in dissimilarity between rate maps of these added wall sessions and the original boundary template (Normalized boundary EMD score: R1, 1.0 ± 0, W1, 1.178 ± 0.03, W2, 1.223 ± 0.03, R2, 1.016 ± 0.01; Friedman test: $X^2$(3)=77.9, p=8.6 × 10$^{-17}$; Post-hoc Wilcoxon signed-rank test: R1-W1, z = −5.35, p=9.0 × 10$^{-8}$, R1-W2, z = −5.58, p=2.37 × 10$^{-8}$, R1-R2, z = −1.27, p=0.20; Bonferroni-corrected α = 0.017; *Figure 2E*). In contrast, the dissimilarity between the same rate maps and an 'added wall' template decreased

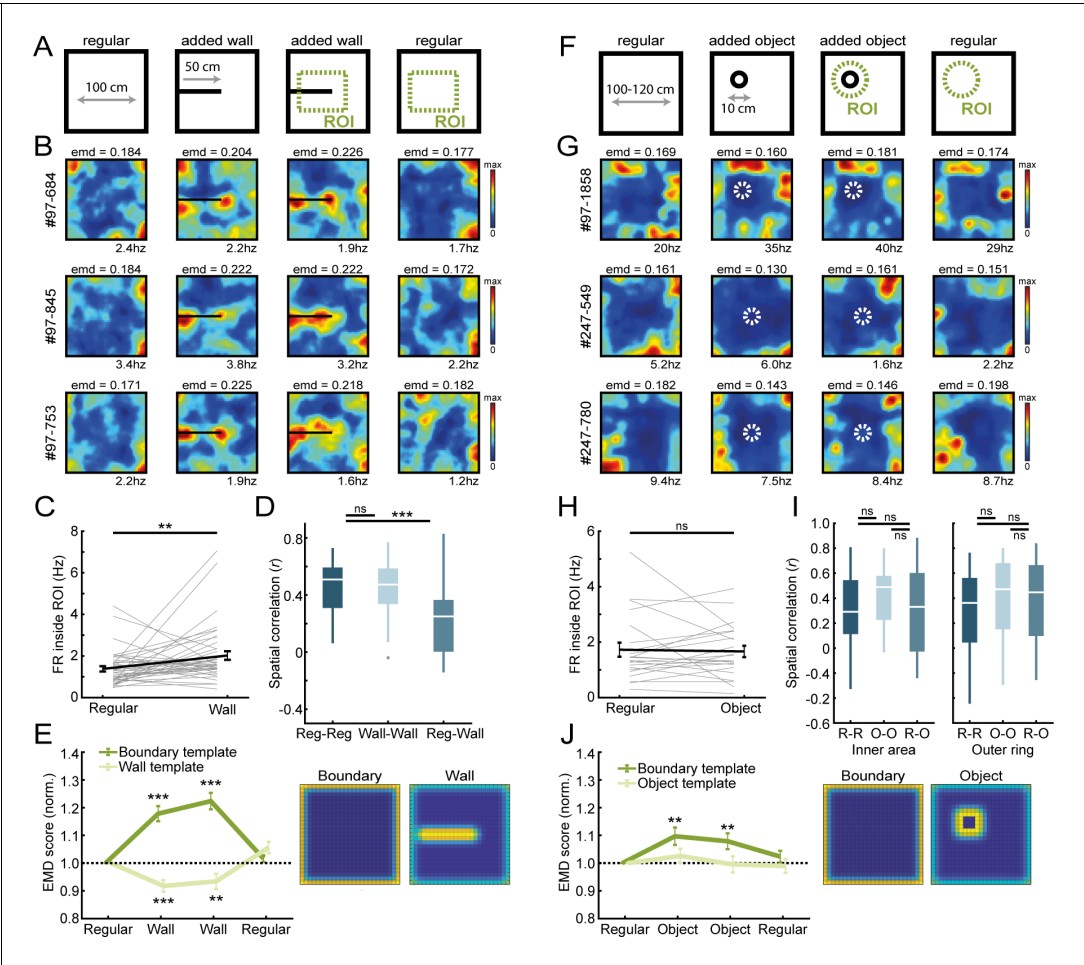

**Figure 2.** Border cells respond to new walls but not to the addition of new objects. (A) An additional temporary wall was placed on the maze in the middle sessions. (B) Spatial rate maps of three typical border cells across regular and added wall sessions during one recording day. (C) Border cells formed new firing fields nearby the added wall, as cells significantly increased their firing rate in the region-of-interest (ROI) area around the central wall. (D) Spatial correlations between rate maps of regular and wall sessions were decreased but remained high within session type. (E) Dissimilarity increased for the boundary template as cells formed fields around the added wall, whereas dissimilarity decreased for an added wall template. (F) A new object was introduced either in the north-west corner or the center of the maze. (G) Spatial rate maps of three example border cells across regular and object sessions. (H) Firing rate of border cells in a ROI around the added object remained unchanged between session types. (I) No significant drop was observed in spatial correlations of border cells by the addition of the object. Correlations were split between an outer ring (four rows) and the remaining inner area to isolate activity related to the outer walls versus the object. (J) There was an increase in EMD scores for the boundary template, whereas the spatial rate maps did not fit with the object template either, as their EMD scores remained unchanged. **p<0.01, ***p<0.001, Wilcoxon signed-rank test.

significantly (Normalized wall EMD score: R1, $1.0 \pm 0$, W1, $0.918 \pm 0.02$, W2, $0.934 \pm 0.03$, R2, $1.057 \pm 0.02$; Friedman test: $X^2(3)=33.7$, $p=2.3 \times 10^{-7}$; Post-hoc Wilcoxon signed-rank test: R1-W1, $z = -3.89$, $p=9.8 \times 10^{-5}$, R1-W2, $z = 2.59$, $p=0.0095$, R1-R2, $z = -2.22$, $p=0.027$; Bonferroni-corrected $\alpha = 0.017$; *Figure 2E*), indicating that firing fields of border cells incorporated boundary information of the added wall. Recordings performed in arenas with a hexagonal, circular, or triangular shape further confirmed the adaptation of firing fields of RSC border cells to the spatial layout of boundaries (*Figure 1—figure supplement 4A–C*).

To investigate generalization to other environmental features, we further added additional objects to the arena and tested the specificity of border responses to the spatial layout (*Figure 2F, G*). The size of the object was 10 cm in diameter, and the animal could walk around or climb on top

without it impeding the animal's movement completely. Contrary to an added wall, RSC border cells maintained tuning only to the outer walls and did not fire whenever objects were inside their receptive field (Circular ROI, eight bins in diameter; Regular: FR = 1.39 ± 0.26 Hz; Added object: FR = 1.42 ± 0.21 Hz; Wilcoxon signed-rank test: z = −0.63, p=0.53; n = 23 border cells; *Figure 2H*). EMD analyses however showed a significant increase in dissimilarity to the boundary template in the object sessions (Normalized boundary EMD score: R1, 1.0 ± 0, O1, 1.097 ± 0.031, O2, 1.079 ± 0.028, R2, 1.024 ± 0.021; Friedman test: $X^2$(3)=14.7, p=0.002; Post-hoc Wilcoxon signed-rank test: R1-O1, z = −2.74, p=0.006, R1-O2, z = −2.62, p=0.009, R1-R2, z = −1.00, p=0.32; Bonferroni-corrected α = 0.017; *Figure 2J*), indicating changes in the rate maps by the object. This change was not due to new firing fields around the object, however, because fitting an 'object' template did not lead to a decrease in dissimilarity during object sessions (Normalized object EMD score: R1, 1.0 ± 0, O1, 1.026 ± 0.026, O2, 0.995 ± 0.029, R2, 0.990 ± 0.025; Friedman test: $X^2$(3)=2.32, p=0.51; *Figure 2J*). The maintenance of original firing fields was further confirmed by spatial correlations across session types that did not drop when comparing rate maps between regular and object sessions, neither for the outer rows of pixels adjacent to the walls (Reg-Reg: r = 0.36 ± 0.07, Object-Object: r = 0.47 ± 0.07; z = −0.88, p=0.38; Reg-Object: r = 0.45 ± 0.07; comparing with R-R: z = −1.28, p=0.20; comparing with R-O: z = 1.16, p=0.25; Wilcoxon signed-rank test; Bonferroni-corrected α = 0.017; *Figure 2I*, right panel), nor the inner area surrounding the object (Reg-Reg: r = 0.29 ± 0.07, Object-Object: r = 0.49 ± 0.05: z = −1.58, p=0.11; Reg-Object: r = 0.33 ± 0.07; comparing with R-R: z = −0.49, p=0.63; comparing with R-O: z = 1.16, p=0.25; Wilcoxon signed-rank test; Bonferroni-corrected α = 0.017; *Figure 2I*, left panel). Taken together, these results indicate that RSC border cells encode information that is specific to boundaries of the spatial layout where cell responses differentiate between the types of added features.

## Border cells retain their tuning in darkness and are not driven directly by whisker sensation

One way for border cells to compute information of boundaries is through direct sensory detection of the walls, for example, by whisking or visual observation (*Raudies and Hasselmo, 2012*). We investigated the importance of direct sensory input on border tuning by removing either visual or somatosensory information of the boundary (*Figure 3A,E*). First, to assess the impact of visual information, we introduced an infrared position tracking system as opposed to regular light-emitting diodes (LED; see Materials and methods) to ensure no visible light was present in the maze for animals to identify boundaries. We recorded the activity of RSC border cells in both dim-light and darkness conditions, but observed no significant changes in EMD dissimilarity scores across the sessions (Boundary EMD score: R1, 0.183 ± 0.001, D1, 0.185 ± 0.003, D2, 0.177 ± 0.003, R2, 0.182 ± 0.002; Friedman test: $X^2$(3)=1.23, p=0.75; n = 21 border cells; *Figure 3B,D*). There were also no changes across spatial correlations between different session types (Reg-Reg: r = 0.42 ± 0.03, Reg-Dark: r = 0.38 ± 0.03; Wilcoxon signed-rank test z = 0.61, p=0.54; Dark-Dark: r = 0.42 ± 0.05, Wilcoxon signed-rank test with Reg-Reg correlation, z = 1.20, p=0.23; Bonferroni-corrected α = 0.025; *Figure 3C*), indicating that activity is not generated through visual sensory input.

In order to examine the role of tactile sensation on boundary representations, we next removed all four outer walls that left a drop-edge above the floor, limiting movement of the animal in the absence of direct somatosensory information of a physical barrier (*Figure 3E*). The EMD scores showed that the majority of border cells were unaffected by the wall removal, with no significant changes in the drop-edge session (Unaffected cells: boundary EMD score: R1, 0.183 ± 0.002, Drop, 0.186 ± 0.003, R2, 0.183 ± 0.001; Friedman test: $X^2$(2)=4.59, p=0.10; n = 17 border cells; *Figure 3G*). However, a subset of cells had disrupted firing nearby the boundary edges (n = 8/25 affected cells, separated based on an increase in EMD that exceeded the 95th-percentile of a null distribution of change, computed using the differences between first and last regular sessions; boundary EMD score: R1, 0.169 ± 0.008, Drop, 0.202 ± 0.004, R2, 0.175 ± 0.006; Friedman test: $X^2$(2)=11.0, p=0.004; Post-hoc Wilcoxon signed-rank test: R1-Drop, z = −2.95, p=0.003, R1-R2, z = 0, p=1.0; Bonferroni-corrected α = 0.025; *Figure 3G*). A similar result emerged from the spatial correlations, where rate map correlations remained high when comparing regular and drop-edge sessions, but only for the unaffected cells that had stable EMD scores across session type (Unaffected cells: Reg-Reg, r = 0.50 ± 0.04, Reg-Drop, r = 0.45 ± 0.04; Wilcoxon signed-rank test:

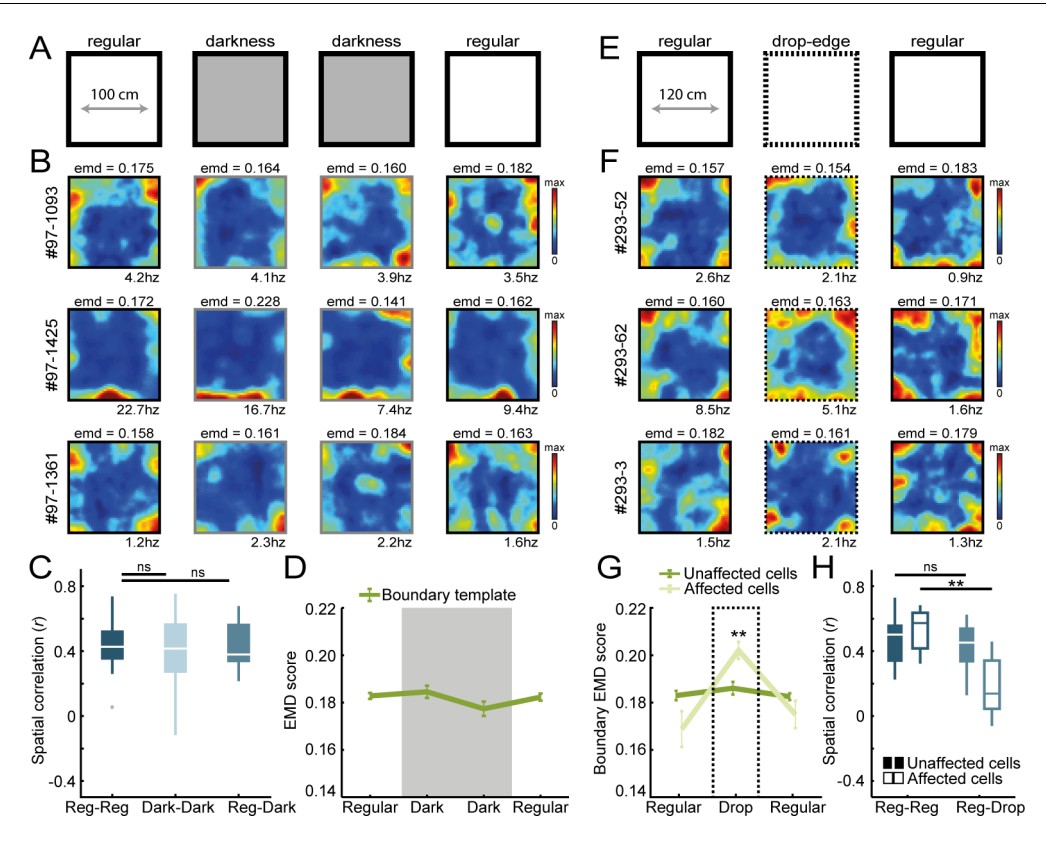

**Figure 3.** Border coding is maintained in darkness and in the absence of physical walls. (**A**) Recordings were performed under no visible light in the middle sessions, and the animal's position was tracked in the infrared spectrum. (**B**) Spatial rate maps of three typical border cells recorded in light and dark conditions. (**C**) Spatial correlations between rate maps of regular and dark sessions remained high, indicating border cells still fired nearby boundaries in darkness. (**D**) There were no changes in EMD scores with the boundary template, confirming that cells maintained their tuning to the outer walls without direct visual detection. (**E**) All four outer walls were removed, leaving only a drop-edge to confine the arena. (**F**) Spatial rate maps of three example border cells across recording sessions. (**G**) Spatial rate maps were maintained for a majority of border cells (unaffected) but a subset of neurons showed disrupted firing near the drop-edge that resulted in an increase in the EMD score on the boundary template (affected). (**H**) Spatial correlations between rate maps of regular and drop-edge sessions remained high for the unaffected cells but decreased significantly for the affected cells. **p<0.01, Wilcoxon signed-rank test.

The online version of this article includes the following figure supplement(s) for figure 3:

**Figure supplement 1.** Control experiments to assess the role of somatosensory information in computing border information.

z = 1.21, p=0.23; Affected cells: Reg-Reg, r = 0.57 ± 0.05, Reg-Drop, r = 0.14 ± 0.07; Wilcoxon signed-rank test: p=0.008; *Figure 3H*).

To directly test the necessity of whisker-mediated tactile sensation, we trimmed the animal's whiskers after which they freely explored the arena (*Figure 3—figure supplement 1A,B*). We did not observe a significant change in the proportion of border cells in RSC after the trimming of whiskers (intact whiskers: 29/276 cells classified as border cells, boundary EMD score = 0.183 ± 0.002; trimmed whiskers: 23/285 cells classified as border cells, boundary EMD score = 0.186 ± 0.005; change in proportion of border cells: z = 1.00, p=0.32, binomial test; *Figure 3—figure supplement 1B*), nor did we see a change in the overall firing rates (before: FR = 1.58 ± 0.38; after: FR = 1.16 ± 0.47; Wilcoxon signed-rank test: z = 1.04, p=0.30; *Figure 3—*

*figure supplement 1C*), suggesting that whisker sensation is not essential for boundary representation. We further recorded from cells located in the barrel field region of the somatosensory cortex (S1bf) in order to understand the nature of somatosensory information through whiskers (*Figure 3—figure supplement 1D–F*). Although we were able to identify a subpopulation of cells that fired nearby boundaries in the somatosensory cortex (n = 23/173 cells classified as border cells using the same criteria; *Figure 3—figure supplement 1G*), an important difference with RSC cells is that S1bf neurons fired consistently near added objects when introduced into the arena, highlighting the selective tuning of RSC border cells to boundaries but not objects (Normalized boundary EMD score: R1, 1.0 ± 0, O1, 1.067 ± 0.01, O2, 1.084 ± 0.01, R2, 0.971 ± 0.01; Friedman test: $X^2(3)=47.1$, p=3.3 × $10^{-10}$; Post-hoc Wilcoxon signed-rank test: R1-O1, z = −3.65, p=2.6 × $10^{-4}$, R1-O2, z = −3.89, p=9.9 × $10^{-5}$, R1-R2, z = 2.95, p=0.003; Bonferroni-corrected α = 0.017; Normalized object EMD score: R1, 1.0 ± 0 O1, 0.977 ± 0.005, O2, 0.967 ± 0.005, R2, 1.015 ± 0.004; Friedman test: $X^2(3)=47.5$, p=2.7 × $10^{-10}$; Post-hoc Wilcoxon signed-rank test: R1-O1, z = 3.50, p=4.7 × $10^{-4}$, R1-O2, z = 3.80, p=1.4 × $10^{-4}$, R1-R2, z = −3.16, p=0.002; Bonferroni-corrected α = 0.017; *Figure 3—figure supplement 1H*). Together, these results suggest that the activity of RSC border cells is not simply driven by the detection of boundaries through visual or tactile sensation, and their distinct firing around boundaries but not objects implies additional computations in the brain.

## Egocentric border cells are invariant following the rotations of allocentric place and head-direction cells

Our results so far suggest that RSC border cells are sensitive to the spatial layout of the environment, allowing for the distinction between boundaries and objects. However, recent reports point to the egocentric tuning of neurons in the postrhinal cortex, RSC or striatum to walls or the center of the maze, suggesting anchoring to local features of the environment (*Alexander et al., 2020*; *Hinman et al., 2019*; *LaChance et al., 2019*). To better clarify how border cells are anchored to the environment, we explored the impact on RSC cells of global environmental manipulations under which allocentric spatial cells shift their firing fields (*Knierim and Rao, 2003*).

We first confirmed that RSC border cells described here have a similar direction tuning as reported in *Alexander et al., 2020*, where spikes that occur near the wall are constraint by specific directions of the animal relative to the boundary (*Figure 4A*). Across the population of border cells identified with the EMD metric, 190 out of 485 neurons (39.2%) had significant egocentric directional tuning, with mean vector lengths above the 95th percentile of a spike-shuffled distribution in all regular sessions (*Figure 4—figure supplement 1A,B*). Conversely, only 190 out of 666 directionally-tuned cells (28.5%) had additional boundary-distance tuning (*Figure 4—figure supplement 1A*), making the population of neurons reported here considerably different from *Alexander et al., 2020*. Projecting trajectory data onto new body-centric axes, where coordinates indicate distance and direction of the nearest wall relative to the animal, confirmed that directionally-tuned border cells fired predominantly whenever the wall occupied proximal space at a particular angle from the animal's viewpoint (*Figure 4B,C*, *Figure 4—figure supplement 2*).

We then sought to establish whether this egocentric constraint was imposed by the head-direction signal through the integration with spatial or sensory cues of the environment, as RSC receives inputs from the anterior limbic system that is a major source of head-direction signals, and a subpopulation of RSC cells are tuned to allocentric head-direction (*Chen et al., 1994*; *Mitchell et al., 2018*). If the egocentric boundary representation of RSC border cells is driven by internally generated global direction signals, realignment of the head-direction cells may affect the preferred tuning direction of RSC border cells. In order to manipulate the tuning of head-direction cells, four blue landmark LEDs were placed on one side of the maze while all other sensory cues were kept invariant across the environment. The entire experimental setup was then rotated 90° clockwise in the middle sessions (*Figure 4D*). As a result, all allocentric head-direction (HD) cells rotated their tuning curves accordingly, although not a full 90° (A-A': median shift = 2.6°, z = 1.23, p=0.23; B1-B2: median shift = 0.8°, z = 0.61, p=0.54; A-B1: median shift = 62.9°, z = 4.62, p=3.8 × $10^{-6}$; A-B1 rotated: median shift = −27.3°, z = −3.07, p=0.002; Wilcoxon signed-rank test; Bonferroni-corrected α = 0.013; n = 28 HD cells; *Figure 4G*). The preferred direction of all border cells that had significant directional tuning, in contrast, remained unchanged (two representative cells in *Figure 4E,F*; A-A': median shift = 0°, z = 0.14, p=0.89; B1-B2: median shift = 0°, z = −1.21, p=0.22; A-B1: median shift = 0°, z = 2.42, p=0.015; A-B1 rotated: median shift = −68°, z = −3.74, p=1.8 × $10^{-4}$; Wilcoxon

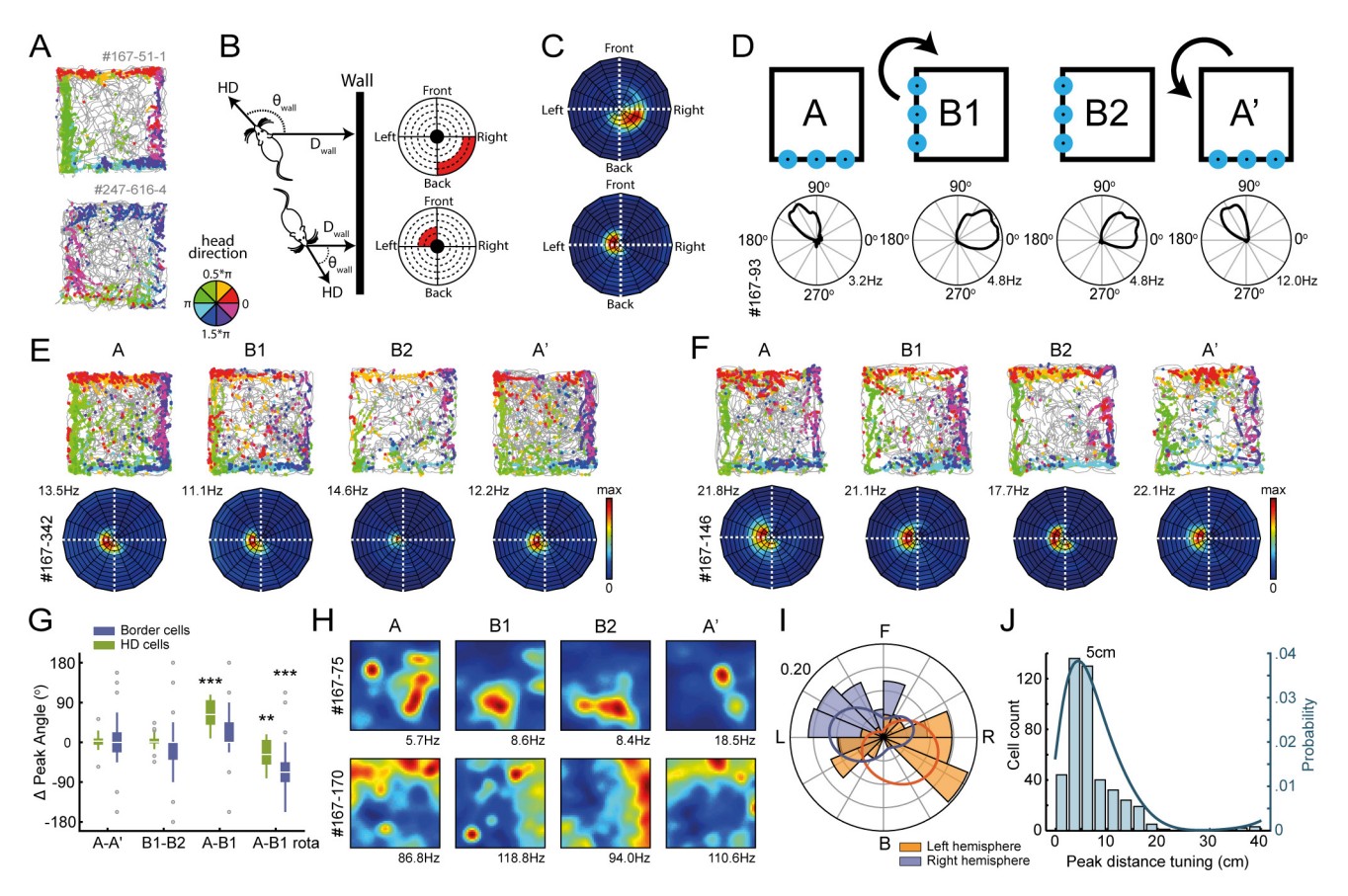

**Figure 4.** Egocentric tuning of RSC border cells has a hemispheric bias and is invariant to the rotation of place and head-direction signals. (**A**) Example trajectory spike plots with spike locations color-coded according to the head-direction of the animal. Most spikes alongside a wall occur only when the animal was in a narrow range of directions. Top: recorded in the right hemisphere. Bottom: recorded in the left hemisphere. (**B**) Trajectory data was projected onto new body-centric border maps, where coordinates indicate the distance ($D_{wall}$) and direction ($\theta_{wall}$) of the closest wall relative to the animal's position and head-direction (HD), respectively. (**C**) Rate maps in this border space for the same example cells shown in (**A**). (**D**) Top: Blue LEDs were placed as distal cues on one side of the maze, and the entire experimental set-up was rotated 90° clockwise in the middle sessions. Bottom: example allocentric HD cell showing its tuning shifted accordingly. (**E-F**) Two example border cells with trajectory spike plots and border rate maps showing egocentric border tuning was stable across rotation sessions. (**G**) Comparison of shifts in direction tuning for allocentric head-direction cells and egocentric border cells across the different sessions, where B1-rota is a rotated version of the rate map in the opposite direction of the physical rotation of the arena, matching the layout again as in A. (**H**) Two examples of spatially-stable cells, defined as having spatial correlations above the 99th percentile of a time-shuffled distribution, which rotated along with the cue. (**I**) Preferred directional tuning of all recorded border cells with significant directional tuning, split according to the location of the electrode in either the left or right hemisphere. (**J**) Preferred distance tuning of all RSC border cells. **p<0.01, ***p<0.001, Wilcoxon signed-rank test.

The online version of this article includes the following figure supplement(s) for figure 4:

**Figure supplement 1.** Directional and distance tuning properties of RSC border cells.

**Figure supplement 2.** Neuron cluster properties along behavioral axes.

signed-rank test; Bonferroni-corrected α = 0.013; n = 31 directionally-tuned border cells; *Figure 4G*). This result indicates that the direction tuning of RSC border cells is generated either through the local sensation of walls independent of the head-direction signal or by the integration of allocentric boundary-position and head-direction coding that rotated together. The RSC's ability to discriminate between boundaries and objects, together with the invariance of border tuning in the absence of visual or tactile signals, goes against the mechanism of local sensation. By contrast, in accordance with the latter possibility, we found that RSC neurons with position-selective firing,

defined as cells with spatial correlations above the 99th percentile of a spike-shuffled distribution that are not border cells, rotated along with head-direction cells in RSC (two example cells in *Figure 4H*; spatial correlations: A-A', r = 0.63 ± 0.018, A-B, r = 0.48 ± 0.058, Wilcoxon signed-rank test: z = 3.52, p=4.4 × $10^{-4}$; A-B rotated, r = 0.70 ± 0.033, Wilcoxon signed-rank test: z = −1.07, p=0.29; n = 32/384 spatially stable cells). RSC border cells thus maintained wall-tuning as their conjunctive coding of position and head-direction rotated together with the environment.

## RSC cells have biased directional tuning to boundaries in the contralateral side of the recorded hemisphere

To further obtain the functional implications, we asked if any directional bias of egocentric tuning exists in RSC border cells by performing large-scale recordings from both hemispheres. Across the population, cells were tuned predominantly to the very near proximity (main peak at 5 cm; *Figure 4J*), although some cells had fields at extended distances up to 20 cm away from the wall. In order to account for a potential proximity bias of our boundary template, we simulated a set of synthetic neurons that fire at specific wall distances using all behavioral data, and found that our boundary template was able to classify cells with firing fields up to 18 cm away from the walls (*Figure 4—figure supplement 1C*). However, cell classification using new templates with fields at increasing distances did not yield a substantial new number of cells (*Figure 4—figure supplement 1D*), confirming that the majority of RSC border cells exhibit distance tuning at the proximity of walls.

Regarding their preferred egocentric tuning direction, we observed a disproportionately biased distribution of preferred directions, dependent on the hemisphere where cells were recorded (Left hemisphere: mean direction = −114°, z = 3.16, p=0.041; n = 41 directionally-tuned border cells; Right hemisphere: mean direction = 41°, z = 38.8, p=9.1 × $10^{-19}$; n = 149 directionally-tuned border cells; Rayleigh test; comparing both distributions: two-sample Kuiper test, k = 3.1 × $10^{3}$, p=0.001; *Figure 4I*). The majority of border cells were tuned to the contralateral side of the recorded hemisphere, although not exclusively (*Figure 4I*). This hemisphere-specific tuning bias implies that boundary representations in RSC may be generated by direct sensory signals, or reflect the command of motor actions, in both of which it arises along the right-left body axis. However, a loss of tactile sensation by whisker trimming had no effect on the extent of directional tuning of border cells (before trimming: 7/49 cells had significant directional tuning, MVL = 0.538 ± 0.06; after trimming: 10/49 cells were significantly tuned, MVL = 0.502 ± 0.08; change in MVL: t(6) = 1.80, p=0.12, t-test; change in cell proportion: z = 1.05, p=0.29, binomial test; *Figure 3—figure supplement 1A,B*), nor did recording in complete darkness affect the directional tuning of cells (light conditions: 5/21 cells had significant directional tuning, MVL = 0.326 ± 0.07; darkness: 4/5 cells maintained their tuning, 2/16 cells were tuned only in darkness, MVL = 0.318 ± 0.05; change in MVL: t(4) = −0.51, p=0.64, t-test; change in cell proportion: z = 0.33, p=0.75, binomial test), implying that this bias is not a direct consequence of the lateralized nature of sensory input.

## Inhibition of MEC input disrupts border coding in RSC but not vice versa

While our results suggest that egocentric boundary coding in RSC is likely formed by using allocentric position and head-direction signals, the exact underlying circuit mechanism has not been determined. The RSC is known to have direct, bi-directional connections with the medial entorhinal cortex (MEC) (*Jones and Witter, 2007*; *Ohara et al., 2018*), which contains several types of neurons that exhibit allocentric spatial tuning such as grid cells, head-direction cells or border cells (*Boccara et al., 2010*; *Hafting et al., 2005*; *Sargolini et al., 2006*; *Solstad et al., 2008*). Given the presence of boundary-responsive cells in both RSC and MEC, it is crucial to establish the direction and extent of functional dependence between these brain regions.

We first addressed the question of whether there is any communication between MEC and RSC in terms of encoding border information using pharmacogenetic inactivation techniques (*Armbruster et al., 2007*). We injected an AAV encoding the inhibitory DREADDs hM4Di into MEC, while simultaneously implanting a 28-tetrode hyperdrive into RSC (*Figure 5A*, *Figure 5—figure supplement 1*). Subcutaneous administration of agonist-21 (DREADDs agonist; *Thompson et al., 2018*) resulted in a drastic reduction of firing after 20 min for a group of cells in MEC that were infected with the virus (26 out of 44 cells recorded near the injection site with additional tetrodes in MEC

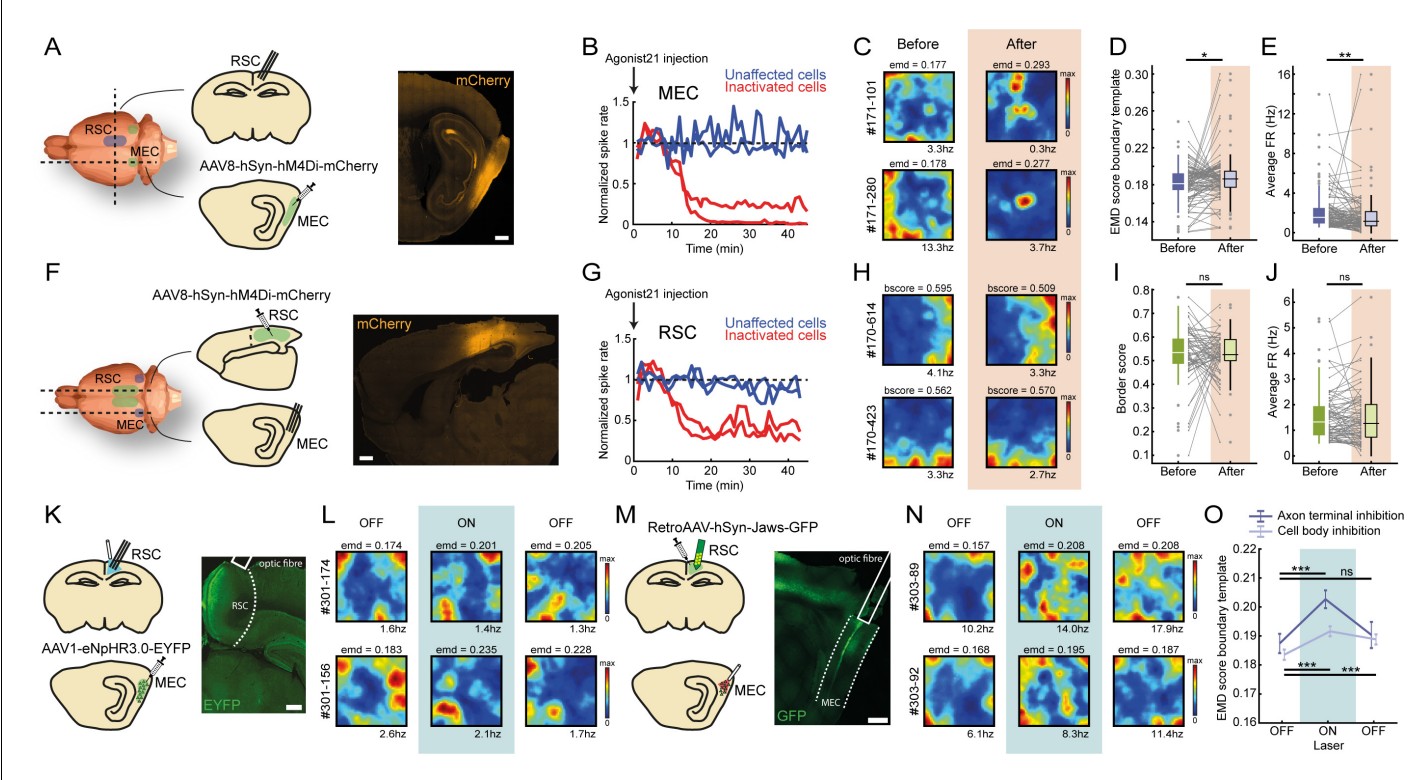

**Figure 5.** Sharp boundary tuning of RSC border cells relies on input from MEC. (**A**) An AAV encoding inhibitory DREADDs hM4Di was injected into MEC while tetrodes were positioned in RSC. Scale bar, 1 mm. (**B**) Four tetrodes were placed locally near the virus injection site, showing a subset of affected MEC neurons that decreased their activity 10–15 min after subcutaneous administration of agonist-21 (DREADDs agonist). (**C**) Two example RSC border cells that were affected by MEC inhibition and lost their spatial tuning. (**D-E**) Border cells in RSC exhibited increased EMD scores as well as lower firing rates after inhibition of MEC. Gray lines indicate individual cells. (**F**) Reversed experiment, with electrophysiological recordings in MEC while the AAV was injected into RSC. Scale bar, 1 mm. (**G**) A subset of RSC neurons decreased their activity after the administration of agonist-21. (**H**) Two example MEC border cells that were unaffected by inhibition of RSC. (**I-J**) Border cells in MEC did not show any significant qualitative changes in border tuning or firing rates after RSC inhibition. Gray lines indicate individual cells. (**K**) An AAV encoding inhibitory Halorhodopsin channels was injected into MEC, while an optrode with tetrodes and optic fiber was implanted in RSC to inhibit MEC axons. Scale bar, 1 mm. (**L**) Two example border cells in RSC that showed disrupted border tuning during inhibition of MEC terminals near the recording site. (**M**) A retroAAV encoding red-shifted inhibitory Cruxhalorhodopsin (Jaws) channels was injected into RSC together with the implantation of a 64-channel silicon-probe, while an optic fiber was placed at the dorsal edge of MEC for the silencing of cells that project to RSC. Histology shows the expression of the virus in neurons located in deep layers of MEC. Scale bar, 1 mm. (**N**) Two examples of RSC border cells that were disrupted due to optogenetic silencing of RSC-projecting neurons in MEC. (**O**) RSC border cells showed disrupted border tuning on a population-level as a direct result of MEC inhibition, both when silencing RSC-projecting neurons in MEC as well as during local inhibition of their axon terminals near the recording sites in RSC. *p<0.05, **p<0.01, ***p<0.001, Wilcoxon signed-rank test.

The online version of this article includes the following figure supplement(s) for figure 5:

**Figure supplement 1.** Tetrode locations and hM4Di expression in the experiments of DREADDs-mediated inactivation.

**Figure supplement 2.** Recording locations and virus expression in the experiments of optogenetic inactivation.

**Figure supplement 3.** Additional analyses for optogenetic experiments.

**Figure supplement 4.** Boundary coding was stable during the laser application without opsin expression.

significantly reduced their firing rates to 47.2 ± 5.5% of their baseline firing; *Figure 5B*, *Figure 5—figure supplement 1*), while not affecting the running speed of the animal (*Figure 5—figure supplement 1*). Inactivation of MEC led to a subsequent partial disruption of firing in a subset of RSC border cells (*Figure 5C*), worsening border tuning that resulted in higher EMD scores (before: EMD score = 0.181 ± 0.002, after: EMD score = 0.186 ± 0.003; Wilcoxon signed-rank test: z = −2.40, p=0.016; n = 102 border cells; *Figure 5D*) and lower overall firing rates after the manipulation

(before: FR = 1.52 ± 0.20 Hz, after: FR = 1.12 ± 0.24 Hz, Wilcoxon signed-rank test: z = 3.15, p=0.0016; *Figure 5E*). Next, we performed a reversed manipulation, injecting the virus encoding DREADDs hM4Di into RSC while recording neural activity in MEC (*Figure 5F*, *Figure 5—figure supplement 1*). Administration of agonist-21 led to similar decreased activity in RSC for the infected cells (*Figure 5G*), but RSC inhibition had no significant effect on MEC border cell tuning (before: border score = 0.53 ± 0.013, after: border score = 0.53 ± 0.01; Wilcoxon signed-rank test: z = 0.56, p=0.57; n = 83 border cells; *Figure 5H,I*) or average firing rates (before: FR = 1.33 ± 0.11 Hz, after: FR = 1.27 ± 0.12 Hz; Wilcoxon signed-rank test: z = −0.31, p=0.76; *Figure 5J*).

While these DREADDs-mediated manipulation experiments suggest the involvement of MEC signals for border computations in RSC, it is possible that MEC inhibition had only an indirect effect on RSC activity, for example by reducing inputs to other communication partners of RSC such as the subiculum (*Roy et al., 2017*). We thus performed two additional manipulation experiments that directly target the MEC-RSC pathway, using optogenetic techniques. We first injected an AAV encoding inhibitory Halorhodopsin chloride pumps (eNpHR3.0; *Zhang et al., 2007*) into MEC, combined with the implantation of tetrodes and an optic fiber in RSC, allowing for silencing of axon terminals of MEC neurons that project to RSC (*Figure 5K*, *Figure 5—figure supplement 2*). Our second approach was to inject a retroAAV (*Tervo et al., 2016*) encoding red-shifted Cruxhalorhodopsin chloride pumps (Jaws; *Chuong et al., 2014*) into RSC together with a silicon-probe for neural recordings, while an optic fiber was placed at the dorsal edge of MEC to allow for cell-body inhibition of RSC-projecting neurons (*Figure 5M*, *Figure 5—figure supplement 2*). This virus was expressed specifically in the deeper layers of MEC, with a gradient that decayed from dorsal to the ventral regions (*Figure 5M*, *Figure 5—figure supplement 2*). Activation of laser light during behavior of the animal led to a subsequent disruption of border coding in RSC, with cells losing specificity of firing near the edges and forming firing fields in the center of the arena (*Figure 5L,N*), resulting in an increase in EMD dissimilarity scores across the population, both for axon-terminal inhibition (Boundary EMD score: laser $OFF_1$, 0.187 ± 0.003, laser ON, 0.203 ± 0.003, laser $OFF_2$, 0.190 ± 0.005; Friedman test: $X^2(2)$=6.9, p=0.032; Post-hoc Wilcoxon signed-rank test: $OFF_1$-ON, z = −3.45, p=5.7 × $10^{-4}$, $OFF_1$-$OFF_2$, z = −1.55, p=0.12; Bonferroni-corrected α = 0.025; n = 34 border cells; *Figure 5O*, *Figure 5—figure supplement 3B*) and cell-body inhibition (Boundary EMD score: laser $OFF_1$, 0.184 ± 0.002, laser ON, 0.192 ± 0.002, laser $OFF_2$, 0.189 ± 0.002; Friedman test: $X^2(2)$=24.5, p=4.7 × $10^{-6}$; Post-hoc Wilcoxon signed-rank test: $OFF_1$-ON, z = −4.97, p=6.7 × $10^{-7}$, $OFF_1$-$OFF_2$, z = −4.11, p=4.0 × $10^{-5}$; Bonferroni-corrected α = 0.025; n = 141 border cells; *Figure 5O*, *Figure 5—figure supplement 3B*) that partially recovered after turning laser light off. Unlike our DREADD inhibition results, there were no significant changes in the overall firing rates of RSC border cells (Laser ON: normalized FR = 0.94 ± 0.04, Wilcoxon signed-rank test: z = 1.68, p=0.09; laser $OFF_2$: normalized FR = 0.96 ± 0.04, Wilcoxon signed-rank test: z = −1.80, p=0.07; *Figure 5—figure supplement 3A*), and there were no observable changes in the behavior of the animals (*Figure 5—figure supplement 3D–F*). Cell classification criteria allowed cells to drop their average firing rates below the rate threshold of 0.5 Hz in the manipulated session, as a reduction in spiking rate indicates disrupted coding. In order to confirm that the increased EMD scores during MEC inhibition were not driven by spurious rate maps of low-firing cells, we repeated the EMD analysis after excluding all cells with an average firing rate below 0.5 Hz in the laser ON session (axon-terminal inhibition, identical results with 34/34 cells; cell-body inhibition, Boundary EMD score: laser $OFF_1$, 0.183 ± 0.002, laser ON, 0.192 ± 0.002, laser $OFF_2$, 0.189 ± 0.002; Friedman test: $X^2(2)$=26.9, p=1.4 × $10^{-6}$; Post-hoc Wilcoxon signed-rank test: $OFF_1$-ON, z = −5.26, p=1.4 × $10^{-7}$, $OFF_1$-$OFF_2$, z = −4.51, p=6.6 × $10^{-6}$; Bonferroni-corrected α = 0.025; n = 138/141 border cells), but obtained the same conclusions.

In order to control for non-specific effects of laser application, we performed an additional control experiment in two animals (*Figure 5—figure supplement 4*). Recording of RSC neurons in the absence of inhibitory opsin expression showed stable firing patterns for border cells near all boundaries of the environment, before, during and after the application of laser light, with no changes for the neurons in their EMD scores (Boundary EMD score: laser $OFF_1$, 0.173 ± 0.002, laser $ON_1$, 0.176 ± 0.003, laser $ON_2$, 0.174 ± 0.004, laser $OFF_2$, 0.170 ± 0.001; Friedman test: $X^2(3)$=5.44, p=0.14; n = 33 border cells; *Figure 5—figure supplement 4D*) or overall firing rates (FR: laser $OFF_1$, 2.95 ± 0.72, laser $ON_1$, 2.43 ± 0.75, laser $ON_2$, 2.39 ± 0.88, laser $OFF_2$, 2.78 ± 0.68; Friedman test:

$X^2(3)=2.64$, p=0.45; *Figure 5—figure supplement 4C*), confirming that the impairment of boundary tuning in RSC during laser application is specific to the silencing of MEC inputs.

In contrast to border cells, our optogenetic manipulations did not affect allocentric head-direction tuning in RSC, with no changes in firing rate (Average FR: laser $OFF_1$, 1.86 ± 0.53, laser ON, 1.68 ± 0.61, laser $OFF_2$, 1.91 ± 0.62; Friedman test: $X^2(2)=0.17$, p=0.92; n = 47 HD cells; *Figure 5—figure supplement 3G*), mean vector length (MVL: laser $OFF_1$, 0.29 ± 0.03, laser ON, 0.26 ± 0.03, laser $OFF_2$, 0.28 ± 0.03; Friedman test: $X^2(2)=6.64$, p=0.036; Post-hoc Wilcoxon signed-rank test: $OFF_1$-ON, z = 2.17, p=0.030, $OFF_1$-$OFF_2$, z = 1.92, p=0.055; Bonferroni-corrected α = 0.025; *Figure 5—figure supplement 3H*), nor a shift in the preferred direction (Shift in preferred direction: laser $OFF_1$-ON, 0.056 ± 0.06; Wilcoxon signed-rank test: z = 0.95, p=0.34; laser $OFF_1$-$OFF_2$, −0.005 ± 0.07; Wilcoxon ranksum test: z = 0.20, p=0.84; Bonferroni-corrected α = 0.025; *Figure 5—figure supplement 3I*) for allocentric head-direction cells, indicating that the head-direction signal in RSC is likely provided by brain regions other than MEC, for example, the anterodorsal thalamic nucleus (*Mitchell et al., 2018*). Further analysis on spatially-stable cells, classified based on high spatial correlations across sessions (see *Figure 4H*), shows that inhibition of MEC did affect firing of spatial cells in RSC, as spatial correlations significantly dropped during laser on sessions (spatial correlations: laser OFF1-OFF2, r = 0.59 ± 0.02, laser OFF1-ON, r = 0.44 ± 0.03; Wilcoxon signed-rank test: z = 5.87, p=$4.5 \times 10^{-9}$; n = 83 spatial cells; *Figure 5—figure supplement 3C*), suggesting that spatial firing beyond border cording is contingent on spatial information coming from MEC. Altogether our manipulation results confirm that both RSC and MEC are involved in a broader border coding network, where border representations in RSC are dependent on direct inputs from MEC but not vice versa.

## RSC border coding is more local and correlated with the animal's future motion

We have shown that MEC input is necessary to maintain sharp border tuning in RSC. However, border cells in both regions differ in their respective firing properties, for example, MEC border cells have firing fields consistently attached to only one or two walls rather than all, indicating allocentric representations of boundaries (two examples shown in *Figure 6A*; variance between average FR near each wall: RSC, CV = 0.103 ± 0.004, MEC, CV = 0.458 ± 0.02; Wilcoxon ranksum test: z = −13.25, p=$4.6 \times 10^{-40}$; *Figure 6—figure supplement 1C*). This raises the question of how spatial information in MEC converges and maps onto RSC border cells. We thus compared the nature and content of information present in spikes of border cells between both regions with respect to the animal's behavior.

We first quantified the spatial information carried by spikes of border cells in RSC and MEC at a population level. The peak firing rates of border cells in RSC were lower than in MEC (RSC: FR = 4.02 ± 0.53 Hz, MEC: FR = 5.30 ± 0.47 Hz, Wilcoxon ranksum test: z = 2.79, p=0.0053; *Figure 6—figure supplement 1D*), but both populations had a similar distribution of peak distance tuning (*Figure 4J*, *Figure 6—figure supplement 1E*). Regarding directional tuning properties, we observed that nearly all directionally-selective border cells in RSC have egocentric directional tuning, while MEC border cells show a high degree of conjunctive selectivity to allocentric head-direction (*Figure 6B*). A decoder based on support vector machines estimated the animal's distance away from the wall using population spiking activity, and performed with high accuracy for both MEC and RSC in the lower distance range (p<0.05 for 0–20 cm, compared with a chance level of 20%; *Figure 6C*), whereas the animal's running speed was not significantly different across distance bins (Kruskal-Wallis test: $X^2(4)=3.24$, p=0.519). However, decoding performance from RSC activity dropped to chance level in the higher distance range (p>0.05 for 30–50 cm; *Figure 6C*), suggesting RSC border cells mainly encode local information. This matches the firing properties of RSC cells which have preferred distance tuning up to 20 cm away from the wall (*Figure 4J*). Conversely, MEC computes distance information that extends toward the center of the arena, with decoding performance above chance-level until the maximum range of 50 cm (p<0.05 for 0–50 cm; *Figure 6C*). Even though MEC border cells fire maximally at the edge of the arena, population vector correlations along neighboring bins decay faster for MEC than RSC, particularly when the animals are more than 20 cm away from the wall (*Figure 6—figure supplement 1A,B*), which allows for MEC cells to distinguish wall distance to a larger extent.

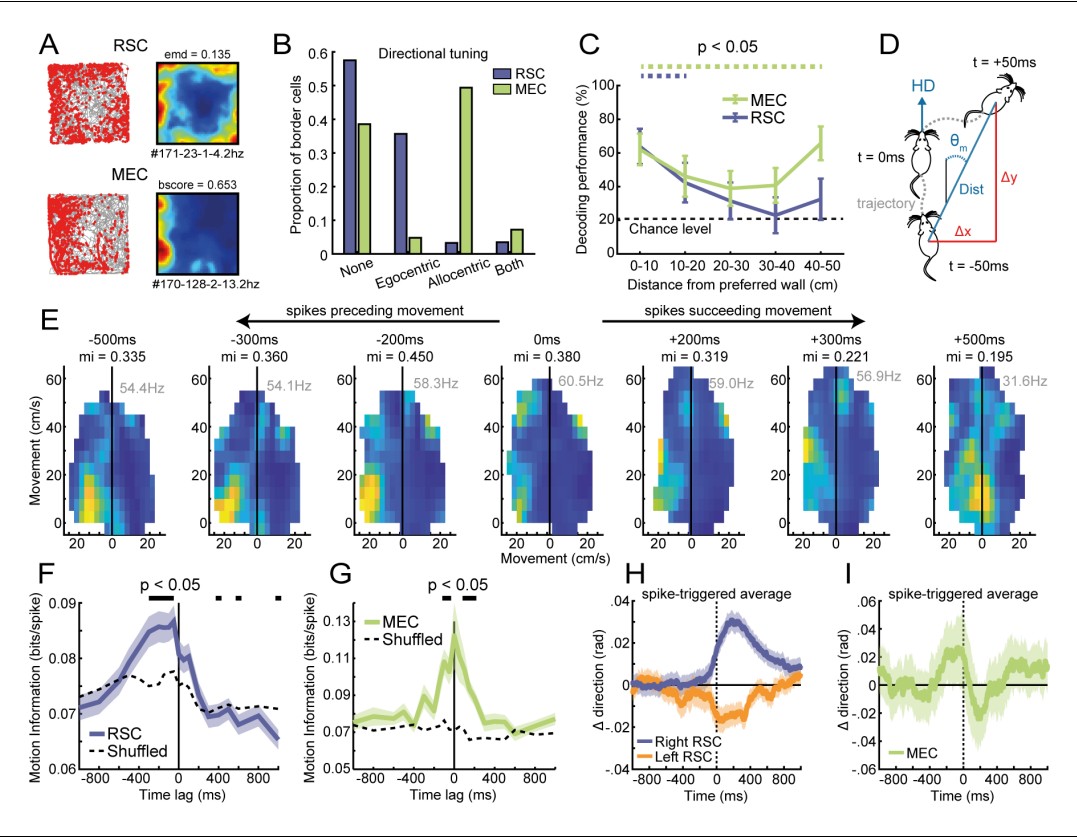

**Figure 6.** Firing of RSC border cells provides local boundary information and is correlated with the animal's future motion. (**A**) Spike trajectory plots and spatial rate maps of typical border cells recorded in RSC and MEC. (**B**) Proportion of border cells in RSC and MEC that had significant directional tuning to allocentric head-direction or egocentric boundary-direction. (**C**) A decoder using a support vector machine (SVM) classifier estimated the animal's distance to the wall based on population spiking activity. Local distance information was present in both regions but extended further into the center of the arena only in MEC. (**D**) Self-motion maps were computed based on short trajectories of the animal, giving lateral and frontal displacements (Δx and Δy, respectively) and distance traveled, *Dist*, in 100 ms time bins relative to the animal's forward head-direction, giving a self-centered moving direction, θₘ, at each timepoint. (**E**) Example motion map of an RSC border cell with spike times shifted in time relative to the animal's motion data. A firing field emerged on left turns when spikes were shifted −200 to −500 ms before motion. (**F**) RSC border cells fired prospective to motion, where the amount of information present in motion maps is maximal when spike timings were shifted −50 to −300 ms earlier. (**G**) MEC border cells by contrast did not show any prospective or retrospective activity. (**H**) Spike-triggered average of changes in direction, calculated as the difference of moving directions in 250 ms bins, where positive values indicate right turns. RSC spikes preceded turning behavior of the animal by ~200 ms, with border cells in opposing hemispheres firing prospectively to ipsilateral turns. (**I**) MEC spikes by contrast were not locked to any change in the animal's behavior. *p<0.05, t-test. The online version of this article includes the following figure supplement(s) for figure 6:

**Figure supplement 1.** Additional population analysis on RSC and MEC border cells.

To further explore if the activity of border cells has behavioral correlates, we finally examined the relationship between cell firing and the animal's self-motion, computing rate maps for movement directions (*Figure 6D*). Shifting spike times in respect to the animal's motion revealed that spikes of RSC border cells tend to precede a particular movement of the animal, as the amount of information present in motion maps is maximal when shifting spikes earlier in time (p<0.05 for the time lag range of −300 to −50 ms compared to shuffled data; *Figure 6E,F*). This shift was not observed in motion rate maps of MEC border cells, which peaks at zero time lag (*Figure 6G*), showing that spike correlations of RSC border cells with prospective motion are not simply due to behavioral restrictions near walls. Next, we aligned the animal's changes in movement direction using the cell's spike timings, which revealed consistent turning behavior of the animal 200 ms after cell firing (*Figure 6H*). The direction of turning was opposite for cells recorded in different hemispheres, where border cells in right RSC fired prospective to right turns, while spikes in left RSC preceded left turns (*Figure 6H*). Such prospective correlates were not observed in MEC border cells (*Figure 6I*), nor in simulated cells

with egocentric border tuning (*Figure 6—figure supplement 1F,G*), confirming the relationship between RSC border cell firing and the animal's next motion. These results together support the idea that RSC and MEC encode different aspects of border representations, playing distinct roles in navigation behavior.

## Discussion

The brain forms boundary representations in two different reference frames, using either egocentric or allocentric coordinate systems. By applying a metric of the earth mover's distance (EMD), we identified a subpopulation of neurons in RSC that increase their firing rates depending on the distance of nearby walls, supporting boundary representations in RSC. These cells are tuned to the distance of all walls of the environment indiscriminately, in contrast to border cells in MEC which fire at the proximity of one or two walls in a particular direction to the room (*Solstad et al., 2008*). We found that firing of RSC border cells is specific to boundaries that impede the movement of animals, while an object introduced into the maze does not elicit a corresponding change of activity nearby. This finding is consistent with the distinction between border and object-vector cells found in MEC, where separate functional cell types encode positional information of both types of features independently (*Høydal et al., 2019*). Furthermore, boundary coding was preserved under no visible light, and a majority of cells maintained their tuning both in the absence of physical walls and the animal's whiskers, which are shared with MEC border cells, as their boundary tuning is also largely maintained without walls present, albeit with some degree of rotational remapping (*Solstad et al., 2008*), and most cells do not form firing fields to an object (*Høydal et al., 2019*). Boundary-vector cells in the subiculum that have reciprocal anatomical connections with RSC (*van Groen and Wyss, 1992*) also possess these same features, including preservation of activity in darkness (*Lever et al., 2009*) and maintenance of boundary tuning without walls present (*Stewart et al., 2014*). RSC border cells, as well as those in MEC and the subiculum, are thus not simply driven by local sensory cues, but likely discriminate boundaries based on a global spatial layout of the environment. Our analyses further revealed that approximately 40% of border cells in RSC have additional egocentric directional tuning toward boundaries, firing predominantly whenever the wall occupies proximal space at a specific angle from the animal's facing direction, unlike MEC border cells that can display conjunctive encoding of allocentric head-direction. This result is consistent with a recent report by *Alexander et al., 2020* which described egocentric boundary vector cells in RSC. Here we demonstrated that this egocentric wall-direction tuning of RSC cells remained invariant under distal-cue rotations, where allocentric position and head-direction signals in RSC rotated together, while silencing of MEC inputs disrupted sharp boundary tuning in RSC, supporting the idea that RSC border cells are formed by conjunctive coding of allocentric boundary-position and head-direction signals that are at least in part derived from MEC.

Anatomically, RSC locates at an interface region of the hippocampus and MEC with sensory and motor cortices (*van Groen and Wyss, 1990*; *van Groen and Wyss, 1992*; *Van Groen and Wyss, 2003*; *Jones and Witter, 2007*; *Sugar et al., 2011*). While both human patients and rodents with lesions in RSC exhibited severe impairment in navigation ability (*Takahashi et al., 1997*; *Vann et al., 2009*), the exact role of RSC has been largely unclear until recently, with several recent studies providing clues for understanding RSC function. An fMRI study in humans demonstrated that RSC is particularly engaged in representing permanent landmarks in the environment (*Auger et al., 2012*), which is consistent with the present finding of border cells as walls can serve as permanent landmarks in an open field arena, especially in the absence of local cues. On the other hand, recording studies in rats have identified several types of spatially-tuned cells in RSC, such as head-direction cells, place cells, and the cells that represent geometric features of the environment (*Alexander and Nitz, 2015*; *Cho and Sharp, 2001*; *Mao et al., 2017*). Because of the existence of these spatially-tuned cells as well as anatomical connections, RSC has been considered an ideal brain region to implement a transformation of spatial representations between egocentric and allocentric coordinate systems (*Bicanski and Burgess, 2018*; *Byrne et al., 2007*; *Mitchell et al., 2018*). The allocentric-egocentric transformation is an essential computational step for navigation because, while spatial representations in the parahippocampal regions about head-direction, places, or borders, are anchored to external features of the environment (i.e., in allocentric coordinates), experiencing the

world through sensory organs and executing motor plans to move through space is referenced to the actor's body and viewpoint (i.e., in egocentric coordinates).

Our findings are in line with the RSC's role in coordinate transformation because both allocentric place and head-direction cells as well as egocentric border cells co-exist in RSC. The question is how such egocentric representation is generated. One possibility is that egocentric border firing is directly driven by sensory perception, such as optic flow or whisker sensation, which is egocentric in nature. This notion is supported by recent reports on self-referenced representations of local space (*Alexander et al., 2020*; *LaChance et al., 2019*), which propose that egocentric representations originate from early cortical and thalamic processing to provide egocentric spatial information to the hippocampus and MEC. However, this possibility is unlikely as our present results show that firing of RSC border cells persisted in the absence of direct visual or tactile detection, and distinguished objects from walls unlike neurons in the barrel cortex. Furthermore, we did not find any significant impact of the silencing of RSC neurons on MEC border cells, showing that RSC inputs are not essential for border coding in MEC. Instead, our results favor the idea that RSC border cells are driven by spatial cells with allocentric tuning. This idea was proposed as a theoretical model (*Byrne et al., 2007*), in which the information about allocentric boundary locations is integrated with head-direction signals to form egocentric border representations. We found that the rotation of head-direction and place cells in RSC, elicited by a cue rotation of the environment, did not affect the egocentric tuning of RSC border cells, indicating that head-direction and position coding in RSC border cells must be bound and rotated together during environmental manipulations, consistent with the proposed circuit model (*Byrne et al., 2007*). Furthermore, by using pharmacogenetic and optogenetic techniques, we found that inactivation of the MEC-RSC pathway resulted in a disruption of position-selective firing of both place cells and border cells in RSC, but not the tuning of head-direction cells. As MEC contains several types of spatially tuned cells in allocentric coordinate frames (*Hafting et al., 2005*; *Moser et al., 2008*; *Sargolini et al., 2006*), our results support the idea that egocentric firing in RSC is formed by the integration of head-direction signals, together with allocentric position information provided by MEC, indicating the transformation from allocentric to egocentric spatial coordinate frames.

Our results, however, also indicate that RSC border cells are not necessarily a simple product of coordinate transformations from MEC cells. The activity of RSC border cells shows a significant bias of tuning direction contra-lateral to the recorded hemisphere, which would indicate that a single hemisphere could transform only half of the potential behavioral space. In addition, the range at which information about wall distance is present is different between MEC and RSC border cells. While RSC border cells provide local information about a nearby wall that is located less than 20 cm from the animal's position, border cells in MEC have extended distance information up to 50 cm (from a wall to the center of the maze). These findings indicate that RSC border cells do not necessarily constitute an egocentric border map as a counterpart of an allocentric map in MEC.

What are the functional implications of a hemisphere bias to boundaries in the animal's contralateral side, if RSC border cells are not directly driven by sensory perception? Our results suggest that this bias is likely a manifestation of the animal's immediate action against the direction of an approaching wall, as movement commands along the left-right body axis are largely lateralized in the brain (*Fritsch and Hitzig, 1870*; *Kim et al., 1993*). Collision detection and avoidance are fundamental roles of sensory-motor systems for many species of animals (*Fotowat and Gabbiani, 2011*), and rodents are also required to detect boundaries to avoid hitting walls or falling off edges. The boundary information in RSC may therefore be used in other brain regions to control the animal's next movements relative to walls or edges. RSC provides inputs to brain regions necessary for motor control and initiation, such as premotor and motor cortices, cingulate cortex, as well as the dorsal striatum (*Guo et al., 2015*; *Jones et al., 2005*; *Yamawaki et al., 2016*). A recent recording study on the dorsomedial striatum has identified a type of neurons that fire near environmental borders in a similar manner as RSC border cells do. However, their egocentric tuning is largely dependent on the animal's movement direction (*Hinman et al., 2019*), rather than head-direction as in RSC border cells (*Alexander et al., 2020*). Notably, our present work discovered that the activity of RSC border cells is also dependent on the animal's movement, but in a prospective manner, exhibiting significant correlations of their firing with the animal's movement direction ~200 ms in the future. This prospective information in RSC may then be transferred to the downstream striatal circuit. We further found that this prospective coding exhibits a similar hemisphere bias as observed in wall-directional tuning,

such that neurons in the right RSC fire prospective to right turns, whereas firing in the left RSC precedes left turns. This lateralized coding scheme may help associate boundary coding with the next appropriate actions, in a way that the right RSC senses a wall to the animal's left, leading to a right turning behavior away from an approaching wall. Our results together thus support the idea that RSC implements coordinate transformation of behaviorally relevant information, pointing to RSC as a key brain region linking the brain's allocentric spatial representations with the animal's behavior.

## Materials and methods

### Subjects

All experiments were approved by the local authorities (RP Darmstadt, protocol F126/1009) in concordance with the European Convention for the Protection of Vertebrate Animals used for Experimental and Other Scientific Purposes. Subjects were 19 male Long-Evans rats weighing 400 to 550 g (aged 3–5 months) at the start of the experiment. Rats were housed individually in Plexiglass cages (45 × 35 × 40 cm; Tecniplast GR1800) and maintained a reversed 12 hr light-dark cycle, with behavioral experiments performed during the dark phase. Animals were mildly food-restricted with unlimited access to water and kept at 85–90% of their free-feeding body-weight throughout the experiment. For recording experiments, eight rats had tetrodes located unilaterally in RSC, either in the left (four rats) or right (four rats) hemisphere. One rat had a 64-channel silicon probe (Buzsaki64-sp; Neuronexus) implanted directly into the barrel field of the right primary somatosensory cortex (S1bf). Four rats were injected with an AAV encoding inhibitory DREADDs bilaterally in either MEC or RSC, combined with a tetrode drive in MEC or RSC in the right hemisphere. For optogenetic inactivation experiments, two rats were injected with a retroAAV (*Tervo et al., 2016*) expressing inhibitory Cruxhalorhodopsin chloride pumps (Jaws; *Chuong et al., 2014*) in the right RSC together with the implantation of a 64-channel silicon probe (Buzsaki64-sp), while an optic fiber was positioned above MEC. Finally, two more rats were injected with an AAV expressing inhibitory Halorhdopsin chloride pumps (eNpHR3.0; *Zhang et al., 2007*) in the right MEC, while eight tetrodes and an optic fiber were implanted together in the right RSC. No statistical method was used to predetermine sample size, although the number of animals used here is similar to previous work.

### Surgery, virus injection, and drive implantation

Anesthesia was induced by isoflurane (5% induction concentration, 0.5–2% maintenance adjusted according to physiological monitoring). For analgesia, Buprenovet (Buprenorphine, 0.06 mg/mL; WdT) was administered by subcutaneous injection, followed by local intracutaneous application of either Bupivacain (Bupivacain hydrochloride, 0.5 mg/mL; Jenapharm) or Ropivacain (Ropivacain hydrochloride, 2 mg/mL; Fresenius Kabi) into the scalp. Rats were subsequently placed in a Kopf stereotaxic frame, and an incision was made in the scalp to expose the skull. After horizontal alignment, several holes were drilled into the skull to place anchor screws, and craniotomies were made for microdrive implantation. The microdrive was fixed to the anchor screws with dental cement, while two screws above the cerebellum were connected to the electrode's ground. All tetrodes were then positioned at 920 µm depth from the cortical surface. All animals received analgesics (Metacam, 2 mg/mL Meloxicam; Boehringer Ingelheim) and antibiotics (Baytril, 25 mg/mL Enrofloxacin; Bayer) for at least 5 d post-surgery.

For tetrode recordings, rats were unilaterally implanted with a hyperdrive that contained 28 individually adjustable tetrodes made from 17 µm polyimide-coated platinum-iridium (90–10%; California Fine Wire; plated with gold to impedances below 150 kΩ at 1 kHz). The tetrode bundle consisted of 30-gauge stainless steel cannulae, soldered together in a 14 × 2 rectangular shape for recordings of the entire RSC, 7 × 4 for anterior RSC, or two squared bundles for bilateral MEC. For RSC, tetrodes were implanted alongside the anteroposterior axis, starting at (AP) −2.5 mm posterior from bregma until −4 mm to −6.5 mm, (ML) 0.8 mm lateral from the midline, (DV) 1.0 mm below the dura, and at a 25° angle in a coronal plane pointing to the midline in order the get underneath the superior sagittal sinus. For MEC, tetrodes were implanted at 4.5 mm lateral of the midline, 0.2 mm anterior to the transverse sinus, at an angle of 15 degrees in a sagittal plane with the tips pointing to the anterior direction. Experiments began at least 1 week post-surgery to allow the animals to recover.

For DREADDs experiments, an AAV8-hSyn-hM4Di-mCherry (a gift from Bryan Roth; Addgene viral prep # 44362-AAV8) was injected with an infusion rate of 100 nL/min using a 10 µL NanoFil syringe and a 33-gauge beveled metal needle (World Precision Instruments). After injection was completed the needle was left in place for 10 min. The virus was injected at two sites for each bilateral MEC (500 nL each at the depth of 2.5 mm and 3.5 mm from the cortical surface, 4.5 mm lateral to the midline, 0.2 mm anterior to the transverse sinus at an angle of 20° in a sagittal plane with the needle pointing to the anterior direction), or four sites along the anteroposterior axis for each bilateral RSC (500 nL each at AP 2.5, 3.5, 4.5, 5.5 mm, 0.8 mm lateral to the midline, at an angle of 25° in a coronal plane pointing to the midline). The flow was controlled with a Micro4 microsyringe pump controller. A small microdrive (Axona Ltd) connected to four-wire tetrodes was additionally implanted nearby the injection site to evaluate the effects of the manipulation. Virus injection was performed in the same surgery as electrode implantation, and recordings began at least 3 weeks post-surgery to allow time for the virus to express.

For optogenetic silencing of MEC terminals in RSC, an AAV1-hSyn-eNpHR3.0-EYFP (a gift from Karl Deisseroth; Addgene viral prep # 26972-AAV1) was injected into right MEC with the same procedure as the DREADDs experiments; injection location was 4.0 mm lateral to the midline, 0.2 mm anterior to the transverse sinus pointing 20° in the anterior direction, with two sites at 2.5 mm and 3.5 mm depths from the cortical surface (500 nL volume each). For optogenetic inhibition of MEC cells projecting to RSC, an AAV-retro-hSyn-Jaws-GFP (a gift from Edward Boyden; Addgene viral prep # 65014-AAVrg) was injected into right RSC at four sites (AP 2.5, 3.5, 4.5, and 5.5 mm, 0.8 mm lateral of the midline and pointing 25° to the midline; 500 nL volume each). Electrode and optic fiber implantation were performed 1 week following virus injection, and experiments began at least 3 weeks post-surgery.

## Spike sorting and cell classification

All main analyses and data processing steps were performed in MatLab (MathWorks). Neural signals were acquired and amplified using two 64-channel RHD2164 headstages (Intan technologies), combined with an OpenEphys acquisition system, sampling data at 15 kHz. Neuronal spikes were detected by passing a digitally band-pass filtered LFP (0.6–6 kHz) through the 'Kilosort' algorithm to isolate individual spikes and assign them to separate clusters based on waveform properties (https://github.com/cortex-lab/KiloSort; *Pachitariu et al., 2016*). Clusters were manually checked and adjusted in autocorrelograms and for waveform characteristics in principal component space to obtain well-isolated single units, discarding any multi-unit or noise clusters. Tetrodes were moved a minimum distance of 80 µm between recording days to find a new set of neurons for the next recording session.

## RSC border cells

We applied a novel template-matching procedure to classify RSC neurons as border cells using the Earth Mover's Distance (EMD), a distance metric from the mathematical theory of optimal transport (*Hitchcock, 1941*; *Rubner et al., 1998*). First, the animal's spatial position occupancy was divided into 4 × 4 cm spatial bins, and the firing rate in each position bin was calculated by dividing the number of spikes with the amount of time spent there. The resulting rate map was smoothed by applying a 2D Gaussian filter (width of 1 bin), and converted to a probability distribution by taking unit weight. We then calculated the Earth Mover's Distance relative to a 'boundary template' using a MatLab implementation of the fastEMD algorithm (https://github.com/dkoslicki/EMDeBruijn, *Koslicki, 2015*; *Pele et al., 2008*; *Pele and Werman, 2009*). This boundary template consisted of a 25 × 25 matrix with each bin's value set to 0, except the outer ring bins with a value of 1, smoothed with the same Gaussian kernel and converted to unit weight. Several additional templates were constructed to assess the effects of behavioral manipulation, adding additional weight in the location of placed objects/walls (*Figure 2E,J*). The EMD distance between a rate map and a template represents the minimal cost that must be paid to transform one distribution into another, with values ranging between zero (identical maps) and one (maximal difference), and is thus a normalized metric of dissimilarity (*Grossberger et al., 2018*).

To assess whether a cell's rate map was significantly similar to the boundary template, we computed a null distribution to compare against using Monte Carlo simulations. We performed 32.000

permutations of a shuffling procedure, and for each iteration we randomly sampled a spike-train from the data, time-shifted this vector along the animal's recorded trajectory by a random interval of at least 4 s and less than the total trial length, wrapping any excess at the and back to the beginning. We then used this shifted data to compute a rate map and calculated the EMD distance relative to the boundary template. Criteria for border cell classification was an EMD dissimilarity score below the 1st percentile of this null distribution in all regular sessions, and an average firing rate of at least 0.5 Hz (*Figure 1D,E*). These cell classification criteria were applied only for the regular sessions of the manipulation experiments.

## MEC border cells

To compare classification results with a related metric, we computed the original border score for each cell (*Solstad et al., 2008*). We first estimated a cell's firing field by isolating a continuous region of at least 200 cm$^2$ and a maximum of 70% of the arena surface where the firing rate was above 30% of the peak firing rate. This was an iterative search until all fields with the above criteria were identified. We next computed the border score, *b*, for each wall separately:

$$b = \frac{c_M - d_M}{c_M + d_M}$$

where $c_m$ was defined as the maximum coverage of any single field over the wall and $d_m$ the mean firing distance, calculated as the average distance to the nearest wall over all bins covered by the field. This was done separately for each of the four walls out of which the maximum score was selected. Cells recorded in MEC were classified as border cells whenever their border score was above the threshold of 0.5 (corresponding to the 99.3th percentile of scores generated from randomly time-shifted spikes) for either of the two recorded sessions, and had an average firing rate of at least 0.5 Hz.

## Head-direction cells

The rat's head-direction was calculated based on the relative x/y-position of two light-emitting diodes (LEDs), corrected for an offset in the placement of the LEDs relative to the animal's true head-direction. For each cell, the mean vector length (MVL) and direction (MVD) was calculated by computing the circular mean and direction from a vector that contained the head-direction of the animal at spike timings in unit space. A cell was classified as a head-direction cell when its MVL was greater than the 95th percentile of a null distribution obtained by thousand-fold Monte Carlo simulations with randomly time-shifted spike trains.

## Border rate maps

Locations of walls were estimated based on the most extreme values of the position of the animal. The animal's distance to the wall was computed for each of the four walls separately by taking the difference between the wall's location and the animal's position in the respective *x* or *y*-dimension, and selecting the lowest value at each time point. The direction of this wall relative to the animal's direction was computed by calculating the angle difference between the animal's true heading direction and a vector pointing directly toward the wall (e.g., relative to an angle of 0° for the east wall, 90° for the north wall). Because 0° corresponds with the 'east' side in angular polar plots, this data was further shifted by 90° to align the front of the animal with the 'north' part in border maps (see *Figure 4C*) to improve visual interpretation of the results.

Firing rate in body-centric border coordinates was calculated by dividing the animal's occupancy in these coordinates into 4 cm distance bins and 20° angle bins. The number of spikes in each bin was then divided by the time spent there, further smoothed using a 2-D Gaussian kernel (one bin width), similar to how spatial rate maps are computed. A cell's preferred direction and distance was obtained by finding the bin with maximal firing rate and selecting the bin's corresponding distance and angle values. For visualization purposes only, this matrix was transformed into a circular diagram shown in *Figure 4*.

To establish the directional tuning of a cell, the wall-direction angle at the time of each spike was taken whenever the animal was located within 20 cm distance of a wall, from which a mean vector length was calculated. This MVL was then compared to a thousand-fold shuffled distribution, where

each iteration produced an MVL value using randomly time-shifted spike timings (similar to head-direction cell classification). If the real MVL exceeded the 95th percentile of this shuffled distribution in all regular sessions, it was considered significantly tuned to wall direction.

## Self-motion maps

First, the animal's movement direction was computed at each time point, using position changes in a 100 ms segment of the preceding and succeeding 50 ms, and calculating the angle of movement by taking the arctangent of the difference in *x/y*-position. The movement directions were then aligned with the animal's forward head-direction, giving moment-to-moment changes in the animal's movement directions from a self-centered perspective (*Ito et al., 2015*; *Whitlock et al., 2012*). The distance traveled in this time bin captures the distance from the origin in self-motion maps, while clockwise or counter-clockwise movements are reflected in shifts over the x-axis. Self-motion data was binned into 3 cm/s bins, and rate maps were computed by dividing the number of spikes by time spent in each bin (*Figure 6E*). For time-lagged analyses, shifted self-motion maps were generated by shifting spike-timing step-wise between −1000 and +1000 ms earlier or later relative to self-motion data. For each time lag, an additional shuffled distribution was computed by shifting the spike-timings a random amount of time, at least 4 s forward, with the excess wrapped around to the beginning, and taking the average over 10 iterations.

From these self-motion rate maps, the total amount of self-motion information could be calculated as:

$$Information = \sum_{i=1}^{N} p_i \frac{\lambda_i}{\lambda} log_2 \frac{\lambda_i}{\lambda}$$

with $i = 1, \ldots, N$ motion bins, $p_i$ the probability of occupancy in bin $i$, $\lambda_i$ the mean firing rate for bin $i$, and $\lambda$ the overall mean firing rate of the neuron (*Skaggs et al., 1996*).

## Decoding analysis

For decoding of wall distance from the activity of border cells in RSC and MEC, the optimal wall with maximum coverage by firing fields was chosen for individual cells (the same procedure as used in border score calculations; *Solstad et al., 2008*). To determine the optimal head-direction to the selected wall for individual border cells, we searched for a range of head-directions (360-degree range in 5-degree steps) that gave the maximum mean firing rate of the cell when the animal was within 20 cm of the wall. We then focused on neural activity when the animal was at this optimal head-direction and in the range of wall distances from 0 to 50 cm at 10 cm steps (five ranges in total), but excluding timepoints where the animal was within 25 cm of other walls to avoid their potential influence. All of the incidents when the animal was in each of the five wall-distance ranges were equally divided into 20 segments in time, and mean firing rates of individual border cells in the 20 segments were assembled across recording sessions. To implement a decoding analysis, 20 cells were randomly chosen, and the order of 20 segments was randomly shuffled for each cell, such that the data in each segment is a collection of firing rates from 20 border cells across various time points of behaviors when the animal was in a particular distance range to the wall. Ensemble firing rates of border cells in one of the segments were selected as a test dataset, and the rest of the data were used to train a support vector machine (using a MATLAB package LibSVM with a linear function; *Chang and Lin, 2011*). Trained weights were then applied to the activity of border cells in the test dataset to estimate the animal's distance to the wall, which was repeated for all segments to be tested (leave-one-out cross-validation), giving a representative decoding performance for the selected population of cells. This procedure was repeated for different cell pairs for 1000 times to estimate a statistical distribution of decoding performance (bootstrap resampling method).

## Behavioral methods

Data was collected over a total of 30–120 min per day while rats foraged for food (chocolate cereal) in a squared open field arena, either 50 × 50 cm, 100 × 100 cm, or 120 × 120 cm in size. Each session consisted of 10–15 min of free exploration in the arena, separated by 5 min of resting time on a pedestal. No curtains surrounded the recording arena, with the exception of the rotation and darkness experiments where all distal cues were blocked completely. The surface of the arena was

elevated 50 cm above the ground, and was enclosed by three black and one white wall with a 50 cm height that were positioned with consistent orientation in the room for all animals. The experimental set-up was extensively cleaned with a 70% ethanol solution in between every recording session to eliminate any odors.

Behavioral manipulation experiments always followed the same protocol of A-B-B-A', where A is a regular session, and the manipulation was performed in B. This allowed for a recovery phase after the manipulation in the final session A'. The only exception was the drop-edge experiment (*Figure 3E*) where the animal had limited motivation; so to ensure good coverage of the arena we reduced the protocol to A-B-A'. All changes to the maze were made in between the first and second session while the animal was resting on a pedestal. For the added wall manipulation (*Figure 2A*), an additional black wall (50 cm length × 50 cm height × 1 cm width) was placed in the maze, protruding from one outer wall at half-length toward the center. For the added object manipulation (*Figure 2F*) either a circular, non-climbable aluminum object (10 cm diameter × 50 cm height) or circular climbable object (10 cm diameter × 10 cm height) was placed in the center, or off-center 40 cm away from the north and west walls.

For the DREADDs-mediated manipulation experiments, animals were injected with agonist-21 (DREADDs agonist 21 dihydrochloride, 3.52 mg/mL [10 mM]; Hellobio) subcutaneously after the first recording session, followed by at least 30 min waiting time to allow the drug to reach the brain and take effect before starting the next recording session. For the experiments using optogenetic methods, laser light was turned on continuously for the duration of the middle session (5 min, with laser power of 20 mW at the fiber tip), after which the animals had at least 5 min of recovery time on the pedestal before starting the final behavioral session. A green laser (532 nm; Shanghai Laser and Optics Century, China) was used to activate eNpHR3.0, while a red laser (632 nm; Shanghai Laser and Optics Century, China) was used to activate red-shifted opsins.

The animal's position and head-direction were obtained by tracking two LEDs on the headstage at 25 Hz and recording under dim light conditions. For darkness sessions, we switched to an infrared OptiTrack camera system (Natural Points Inc) under the assumption that rats have limited vision in the higher wavelengths, with cone sensitivity tapering off rapidly above 600–650 nm (*Jacobs et al., 2001*). Six Flex three cameras were positioned 2 m above the arena surface on a ceiling mount, at a 45–60° angle pointing downwards, that used infra-red illumination (peak spectral emission at 850 nm) to track the location of three reflective markers in an asymmetric frame attached to the headstage. Position and direction data were acquired and processed using Motive 2.0 software. To ensure no visible light was present for the animals, all lights were turned off and small light sources in the room such as computer and sensor lights were taped off, while the arena was enclosed by a thick, black curtain. A room lamp was turned on for dimly light conditions until 10 s before the start of the recording, and turned on again during the inter-trial interval duration of 5 min. During recording, the experimenter remained stationary and silent near the arena throughout the recording while scattering food pellets.

## Histological procedures

Once the experiment was completed, animals were deeply anesthetized by sodium pentobarbital and perfused intracardially with saline, followed by 10% formalin solution. Brains were extracted and fixed in formalin for at least 72 hr at 6° C temperature. Frozen coronal sections were cut (50 μm) and stained using cresyl violet and mounted on glass slides. Electrode tips were identified by comparison across adjacent sections, with the location of recorded cells estimated by backward measurement from the most ventral tip of the tetrode tracks.

## Statistical procedures

All statistical tests were two-sided and non-parametric unless stated otherwise. Error bars in all figures represent the standard error of the mean (SEM). All values mentioned in the text are medians ± SEM.

## Acknowledgements

We thank Martin Vinck for suggesting the approach with the Earth Mover Distance (EMD) and providing initial software for analysis; Diogo Santos-Pata for discussion and comments related to the

manuscript; N Vogt, S Zeissler, E Northrup and G Wexel for animal care; F Bayer and A Umminger for building the behavioral mazes; Robert Gebauer for technical support; and all members of the Ito laboratory for discussions.

## Additional information

### Funding

| Funder | Grant reference number | Author |
|---|---|---|
| Japan Science and Technology Agency | JPMJPR1682 | Hiroshi T Ito |
| H2020 European Research Council | 714642 | Hiroshi T Ito |
| Max-Planck-Gesellschaft | | Hiroshi T Ito |
| Behrens-Weise-Foundation | | Hiroshi T Ito |

The funders had no role in study design, data collection and interpretation, or the decision to submit the work for publication.

### Author contributions

Joeri BG van Wijngaarden, Conceptualization, Data curation, Formal analysis, Validation, Investigation, Visualization, Methodology, Writing - original draft, Writing - review and editing; Susanne S Babl, Data curation, Investigation, Methodology; Hiroshi T Ito, Conceptualization, Data curation, Formal analysis, Supervision, Funding acquisition, Validation, Investigation, Methodology, Writing - original draft, Writing - review and editing

### Author ORCIDs

Joeri BG van Wijngaarden (iD) https://orcid.org/0000-0001-5208-665X
Susanne S Babl (iD) https://orcid.org/0000-0003-4818-2782
Hiroshi T Ito (iD) https://orcid.org/0000-0001-7726-0781

### Ethics

Animal experimentation: The experiments were approved by the local authorities (RP Darmstadt, protocol F126/1009) in concordance with the European Convention for the Protection of Vertebrate Animals used for Experimental and Other Scientific Purposes.

### Decision letter and Author response

Decision letter https://doi.org/10.7554/eLife.59816.sa1
Author response https://doi.org/10.7554/eLife.59816.sa2

## Additional files

### Supplementary files

• Transparent reporting form

### Data availability

Raw data deposited in Dryad Digital repository (https://doi.org/10.5061/dryad.8cz8w9gnj).

The following dataset was generated:

| Author(s) | Year | Dataset title | Dataset URL | Database and Identifier |
|---|---|---|---|---|
| van Wijngaarden JB, Babl SS, Ito HT | 2020 | Entorhinal-retrosplenial circuits for allocentric-egocentric transformation of boundary coding | https://doi.org/10.5061/dryad.8cz8w9gnj | Dryad Digital Repository, 10.5061/dryad.8cz8w9gnj |

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
