## [Decision Letter]

**Acceptance summary:**

This work identifies a population of neurons in retrosplenial cortex that respond to environmental barriers at a range of distances (weighted towards short distances) some of which are also tuned to the egocentric direction of the barrier. Interestingly, a variety of behavioural, opto- and chemo-genetic manipulations indicate that these are not simple sensory responses, but potentially reflect a representation for egocentric action (such as turning away from barriers) constructed from allocentric representations in the hippocampal formation.

**Decision letter after peer review:**

Thank you for submitting your article "Entorhinal-retrosplenial circuits for allocentric-egocentric transformation of boundary coding" for consideration by *eLife*. Your article has been reviewed by three peer reviewers, one of whom is a member of our Board of Reviewing Editors, and the evaluation has been overseen by Laura Colgin as the Senior Editor. The reviewers have opted to remain anonymous.

The reviewers have discussed the reviews with one another and the Reviewing Editor has drafted this decision to help you prepare a revised submission.

We would like to draw your attention to changes in our revision policy that we have made in response to COVID-19 (https://elifesciences.org/articles/57162). Specifically, we are asking editors to accept without delay manuscripts, like yours, that they judge can stand as *eLife* papers without additional data, even if they feel that they would make the manuscript stronger. Thus the revisions requested below only address clarity and presentation, but do include requests for additional analyses.

Summary:

The authors report e-phys recording of neurons in retrosplenial cortex that respond to environmental barriers (walls, inserted barriers and drop-edges) at a range of distances (weighted towards short distances) and with tuning to the egocentric direction of the barrier (although the amount and distribution of directional tuning is slightly unclear, with 185/485 cells exceeding a >99th percentile in shuffled directionality).

By using darkness, changing the nature of the barrier (walls to drop edges) and cutting whiskers, they show that these responses are not simple unimodal sensory responses. By rotating polarising environmental cues, they show that the tuning of head-direction cells and spatially tuned cells in RSC rotate, but the egocentric directional tuning of the RSC barrier-responding cells is retained.

They compare these to border cells in medial Entorhinal ctx., showing that EC border cells have higher firing rates and can be used to decode the distance to the walls for a longer range (up to 50cm) than RSC cells. The RSC cells with directional tuning are lateralized so that those in one hemisphere tend to be tuned to contralateral egocentric direction (similar to the cells reported by Alexander et al.,). They also show that RSC cells (but not EC border cells) reliably precede upcoming movements (turning away from the barrier).

Finally, they perform chemo- and opto-genetic silencing in each region while recording in the other, showing that the RSC cells are affected by EC input but not vice versa. That is, inactivation of EC by DREADDs reducing tuning and firing rates in RSC, and halorhodopsin or jaws inactivation of EC projections to RSC disrupted RSC cell firing near boundaries, shifting firing to the centre of the environment.

They interpret their results in terms of top-down activation of egocentric responses (turning away from barrier) driven by allocentric representations of barriers in MEC.

The paper is interesting and (mostly) clear, potentially showing a sophisticated egocentric representation generated top-down from allocentric representations. However, there are several issues that would need to be clarified or resolved before publication in *eLife*.

Essential revisions:

1) Need for much more cautious interpretation of the MEC inactivation experiments. Is it fair to ascribe such a strong role for MEC on the basis of these data, or might it be one of many potential inputs?

a) Subsection “Inhibition of MEC input disrupts border coding in RSC but not vice versa” Figure 5, and associated Supplementary Figures etc. The DREADD experiment shows powerful reduction of affected MEC cells without affecting running speeds. Nice work. The effects on the RSC cells, though, are rather mild. After MEC inactivation, the average EMD boundary template score was 0.186. Yes, this was lower than before inactivation (before was 0.181, p = 0.016), but the net effect of inactivation is that the average EMD-border score is now 0.186 and thus still well under the 99% classification threshold to be defined as a border cell.

b) The ratemap illustrations of this manipulation, Figure 5C left, show two cells, with EMD values before of 0.177 and 0.178 and after of 0.293 and 0.277. Of over 100 border cells, they show the most unrepresentative cell and the third most unrepresentative cell. Something more representative should be shown.

c) Similar points as a) apply to the other two inactivation experiments using optogenetics. The cell-body inhibition results are particularly weak in effect, with laser ON EMD scores averaging 0.0192. This average is 0.001 above the relatively strict 99% threshold of 0.0191, and 0.003 above the second laser off average. Thus, the cells are on average still very border-like.

d) as with b) Similarly unrepresentative cells it seems are shown for Figure 5L and 5N.

e) The disruption of firing caused by inactivation of the MEC seems slight (Figure 5D,O), and the examples in 5L (and to some extent 5N) are not convincing because the firing patterns do not seem stable across the two 'OFF' trials, so it is hard to be sure that changes in the 'ON' trial are due to the manipulation. To what extent does the laser stimulation (Figure 5O) increase the 'messyness' of firing rather than changing its tuning characteristics – eg reducing spatial information/stability or increasing excitability (are firing rates different)?

f) The chemogenetic and optogenetic manipulations are lacking standard controls.

Specifically, there is no non-DREADDs group or sham injection recordings for the chemogenetic experiment, and there is no control virus group in the optogenetic experiments. As such the effect could be due to systemic effects in the former and with heating in the latter. The DREADDs experiments do have an internal control with the RSC-MEC reversal inactivation, but not the optogenetic experiments. That being said, the cell body inhibition experiment gives more confidence in the result.

g) These findings are interpreted with exaggeration.

Abstract "These egocentric representations…require inputs from MEC." Subsection “Inhibition of MEC input disrupts border coding in RSC but not vice versa” "While these DREADDs-mediated manipulation experiments suggest the necessity of MEC signals for border tuning in RSC…". Figure 5 legend: "RSC border cells require input from MEC to maintain their boundary tuning". Necessity and Require are untenable inferences from the modest effects shown, and this should all be rephrased so casual readers are not misled.

They should perform a sanity-check analysis where cells with peak rates of say 1Hz are excluded from the analyses. If a cell is not really firing, it may not be that informative to examine the spatial features of the few available spikes.

h) Subicular boundary related inputs. Boundary coding being both preserved in darkness (Lever et al., 2009; see also Brotons-Mas et al., 2010), and most cells maintaining their tuning without walls present (Lever et al., 2009; Stewart et al., 2014) is shown in the subiculum and thus there is a source of boundary-coding information additional to the Entorhinal cortex that shares some key features with these retrosplenial border cells. The projection to the retrosplenial cortex from the dorsal subiculum, where boundary vector cells have been found, is dense (see e.g. Wyss and Van Groen, 1992). I think Rosene and van Hoesen, 1977 suggest the main cortical afferent to the granular RSC originates in the subiculum. Thus, consideration of boundary information coming into the RSC should mention such boundary cell and anatomy tracing work.

2) What are the defining characteristics of the RSC 'border cells', are they directionally tuned, how do they relate to other boundary-responsive cells, and what to call them?

a) Quantification of border scores is by comparison to a Gaussian smoothed template of firing at the borders. However, the comparison method (earth mover distance, EMD) is not clear – giving an intuitive explanation, such as the total distance moved by all units of firing rate to match the firing rate and template distributions would be helpful. More intuition for the numbers would be gained by showing cells with values near the classification thresholds, not just at/near the tails. Figure 1F shows values of 0.14, 0.145, 0.159…. and then 0.222 and 0.312. Please show cells near 0.1906 cutoff. Relatedly, in Figure 1E, show the EMD values corresponding to 95 and 90% cutoffs.

b) It is not clear the extent to which spiking has to be restricted to the borders of the environment, how the method captures spiking that is displaced a certain distance from a border, and how the distance and egocentric direction tuning of each cell was found. If the template is only at the border are more distally tuned cells missed?

Is this same measure applied to the MEC (for fair comparison of the MEC and RSC it should be)? And does it find cells that fire distant from the border in MEC? This is particularly relevant given the puzzle that spatial representation occurs up to 50cm from the border by MEC 'border cells'? Did distance tuning differ between RSC and MEC (please show the distance tuning distributions for both areas)?

c) The comparison to actual border cells (that must fire continuously along a border) is important – the new score does not penalise gaps in firing (hence the appearance of a grid cell in Figure 5L top left?), nor does it require an allocentric tuning direction (a characteristic of border cells and boundary vector cells).

How strong is the tuning to egocentric direction or are these cells that mostly fire near a border in any direction? 185/485 egocentrically tuned cells seems low. Do 300/485 have no directional modulation, or is there qualitative egocentric modulation but below the statistical threshold? If not directionally modulated at all they can't be classified as either egocentric or allocentric.

The claims made (see also 2d) warrant further investigation of the potential differences between their confirmed egocentric border cells and the potentially numerous allocentric border cells within the RSC. Please provide the distribution of egocentric (and allocentric) directional tuning strengths across the populations of 'border cells' in RSC and MEC.

d) Clarification in the language used in the Abstract, Introduction, and Discussion section seems vital. The authors make much of the distinction between the allocentric boundary cells in other regions, and the egocentric boundary cells here. Furthermore, the abstract offers the summary: 'Border cells in RSC…are sensitive to the animal's direction to nearby borders'. Is Earth Mover Distance (EMD) alone egocentric? If not, and all of the analyses are on the EMD population, the result should not be framed as allocentric to egocentric transformation. If the egocentric border cells were analyzed throughout, that would justify the title and framing.

It is confusing to refer to both the barrier-responsive cells in RSC and the previously documented EC border cells as simply 'border cells', when the two populations appear to be different. 'Border cells' were defined by Solstad et al., 2008 as cells that fire when the animal is right next to a barrier in a specific allocentric direction (thus distinguishing them from the pre-existing 'boundary vector cells'). They subsequently suggested that border cells respond to direct contact with a physically present barrier, unlike 'object vector cells' which also respond to an object suspended above them (Hoydal et al., 2019), or boundary vector cells which can respond to the previous location of a barrier (Poulter et al., bioRxiv). The barrier-responsive RSC cells are clearly not 'border cells' – they can respond to barriers at a distance, some with a tuning to egocentric direction, and (the authors argue) are not a direct sensory response to the barrier.

What should they be called? If they think the population is generally tuned to egocentric direction (this is not clear, see next point below) 'egocentric boundary vector cells' would be technically correct, but is a bit of a mouthful, the main thing to avoid would be calling them something they are not. Perhaps referring to them by a label containing 'RSC' would at least make clear when they are referring to their RSC cells and when they are referring to border cells in EC ('RSC boundary cells' or similar).

e) Description of MEC border cells is a little misleading. "These features are shared with MEC border cells, as their boundary tuning is also maintained without walls present (Solstad et al., 2008)".

The data of Solstad et al., 2008 (Figure S8 and S9) show rather altered firing when walls are removed. (A) Of 10 border cells studied in the wall-no wall manipulation, only one maintained its field in the no wall environment, the others did complex or seemingly rotational remapping. (B) Furthermore, Solstad S9 suggests that the seeming-rotational remapping of border fields occurred despite the directional firing of simultaneously recorded head direction cells' being stable throughout the wall-no wall manipulation. Thus, even the seemingly simple rotational remapping may be more complex. In all, this section thus needs revision and clarification.

3) 'Complete Darkness'. If a darkness condition disrupts, there is less burden on the experimenter to ensure its completeness; e.g. darkness greatly reducing gridness implies vision aids grid cell firing in mice (Chen et al., 2019) is a safe inference, and if the darkness was not complete, this does not matter that much. Here, that the darkness was without effect is the finding itself, so there is a higher burden on the experimenter to detail this manipulation. Details in subsection “Behavioral methods” are minimal. It is valuable to know, e.g.: how the room has been prepared for 'complete darkness' (5 minutes from light to dark is a very fast transition, does this generic intertrial interval apply also to the dark trials?), something of the previous light-exposure background of the rats before each one of the infra-red trials is conducted, what happens in the inter-trial intervals, LED technicalities, including the IR 'bleed' nm range, not just the peak value (850nm?), brightness settings, height of cameras from floor, and so on, enabling replication of the setup. My understanding from Flex3 details on the Optitrack website is that there will be in total over 150 LEDs shining upon the environment from 6 cameras: the phrase 'complete darkness' seems untenable. After adaptation, the rats may be able to see in this. If it really was just 5 minutes between light and dark there is minimal time for adaptation, that will be fine, though the second dark trial may involve some dark adaptation if darkness persists in the inter-trial interval. Also: What were the experimenters doing? E.g. more stationary in the darkness condition than the light condition? For evidence that humans, rats, mice and cats can see in the supposedly invisible infrared spectrum, if they are dark-adapted, see e.g. Pardue et al., 2001 and Palczewskaet al., 2014. A paper on good ferret vision at 870nm may be of interest (Newbold and King, 2009).

4) Object insertion

This manipulation is not yet convincing.

a) Though it features in their abstract as a main finding, there are not many cells for this experiment.

b) It seems sub-optimal that the ROI around the object is square, not circular, and quite a large square.

c) Perhaps 10 cells increase their firing in this large square ROI, and others decrease their firing. A lack of perturbation by object insertion is not self-evident from these data; rather there could be some cell-specificity, with some cells with firing being actively inhibited by the object, and others being excited by it. There seem to be quite large reductions in firing in 3 cells.

d) Thus, their aggregate analysis is perhaps a little simplistic, and misses out cell-specific responses. As there is not much statistical power, it might be simpler just to show the majority of these cell responses in a supplementary figure.

e) Importantly, the EMD templates are biased towards increasing the likelihood of a null finding. Put simply, the walls in the boundary template have two rows of high firing bins near them, whereas in the object template, these same walls have only one row of high firing bins near them, and moreover this row is of lower rate. In marked contrast, the object in the object template has three rows of higher rate bins around it. (To be clear, the authors mention this bias in their Materials and methods section: "adding additional weight in the location of placed objects/walls".) Thus, the object is expected (by the template) to elicit high firing for a more extended distance and at higher rates than that at the boundary even though the boundary is an extended cue and should influence the cell more. There is no need for such a biased hypothesis.

f) More details should be provided as to the previous experience with the objects. Might there be an inhibition by novelty? g) Object size: Figure 2F says the object was 15cm in diameter, but Figure 2—figure supplement 1, part g and the main text says 10cm. Please check and clarify sizes for all experiments, including the size of the ROI around the object.

In summary, there is no doubt whatsoever that the walls exert a greater influence than the object, but that is different from saying that "firing of RSC border cells…is invariant to an object introduced into the maze" (Discussion section). This is not an accurate summary when even their analysis as it stands shows a substantial increase in EMD to the boundary template in the object condition.

[Editors' note: further revisions were suggested prior to acceptance, as described below.]

Thank you for resubmitting your work entitled "Entorhinal-retrosplenial circuits for allocentric-egocentric transformation of boundary coding" for further consideration by *eLife*. Your revised article has been evaluated by Laura Colgin (Senior Editor) and a Reviewing Editor.

The manuscript has been improved but there are two remaining issues that need to be addressed before acceptance, as outlined below:

1) The statement in the Abstract where it says the cells "depend on inputs from MEC" should be moderated to something like "are influenced by inputs from MEC" to be more consistent with the point made in the reviews.

2) In the response you say, "we decided to include cells that have minimal firing only in opto/chemogenetic manipulation sessions, as this is a clear indication of disrupted firing due to the manipulation." Please confirm the process by which the population of cells was selected on which to test the effect of the manipulation.

---

## [Author Response]

Essential revisions:1) Need for much more cautious interpretation of the MEC inactivation experiments. Is it fair to ascribe such a strong role for MEC on the basis of these data, or might it be one of many potential inputs?

We understand the reviewers’ concerns regarding our results of the inactivation experiments, as only a subset of neurons showed disruption of boundary tuning. Because virus expression is limited to a subpopulation of the cells in MEC, and both pharmacogenetic and optogenetic methods do not necessarily abolish the neuronal firing completely, the methodology does now allow us to distinguish whether such partial disruption is due to partial silencing of MEC, or because of compensatory inputs from other brain regions. Because we observed consistent disruption of RSC border cells with 3 different inactivation approaches (DREADDs, optogenetics soma retrogradely and axon terminals), we concluded that direct inputs from MEC are necessary to maintain sharp boundary tuning in RSC. However, we do not rule out the potential contributions of additional pathways, such as inputs coming from the subiculum. This point will be discussed further in the following points and described in the Discussion section of the revised manuscript.

a) Subsection “Inhibition of MEC input disrupts border coding in RSC but not vice versa” Figure 5, and associated Supp Figuresetc. The DREADD experiment shows powerful reduction of affected MEC cells without affecting running speeds. Nice work. The effects on the RSC cells, though, are rather mild. After MEC inactivation, the average EMD boundary template score was 0.186. Yes, this was lower than before inactivation (before was 0.181, p = 0.016), but the net effect of inactivation is that the average EMD-border score is now 0.186 and thus still well under the 99% classification threshold to be defined as a border cell.

The reviewers are correct in pointing out that the effects of MEC inactivation on RSC boundary template scores are mild on a population level, at least in terms of shifts in population medians described in Figure 5. However, we would like to highlight several considerations in the interpretation of these results:

1) We injected a small volume of the virus so that the expression would be confined to MEC. In the histological sections, we observed a small bias of virus expression along the dorsoventral axis of MEC, with some cells without expression (Figure 5—figure supplement 1). Furthermore, DREADDs-mediated inhibition generally reduces firing rates to only ~50% of baseline and does not abolish activity completely (Armbruster et al., 2007). In our experiments, we found that 59% (26/44) of the recorded cells near the injection site significantly reduced their firing rates to 47.2 ± 5.5% of the baseline after Agonist-21 administration. The activity of MEC cells is thus only partly disrupted by the DREADDs method, and it is possible that the remaining activity of MEC cells is sufficient to maintain some degree of boundary tuning. These points are now described in the legend of the supplemental figure.

2) While tracing studies have shown direct bi-directional connections between RSC and MEC (Jones and Witter, 2007; Ohara et al., 2018), it is unclear what connectivity topology exists between both regions, and RSC border cells may receive inputs from multiple neurons along the dorsoventral axis of MEC. The previously-proposed model indeed suggests that egocentric border tuning is formed by the integration of multiple allocentric border cells (Bryne et al., 2007). This would predict only partial disruption of border tuning after the perturbation of a subpopulation of input cells.

Considering these two points, we expected to observe some border cells in RSC to remain unaffected after a partial inhibition of MEC, making the population’s mean EMD scores by itself not necessarily a good indicator to assess the overall impact of MEC inputs. We further consider the possible involvement of other pathways in maintaining the border tuning of some RSC cells that were not disrupted after the manipulation, but neither pharmaco- or optogenetic methods can distinguish these possibilities. Because of these limitations, we instead sought to determine whether a group of cells show disrupted boundary coding due to inhibited MEC input by examining the change of EMD scores across the population. There were a sufficient number of cells disrupted to cause an overall increase of EMD score in population median, from which we conclude that MEC provides boundary information for RSC to form sharp boundary tuning.

In the revised manuscript, we have added additional control experiments in which the laser was applied without opsin expression, and confirmed the overall stability of EMD scores across consecutive sessions, which supports that the increase of EMD scores (as observed in the DREADDs-mediated manipulation) cannot be explained by intrinsic instability of the boundary tuning of RSC cells.

We discuss this point in subsection “Inhibition of MEC input disrupts border coding in RSC but not vice versa”, and further elaborate on the control experiment in comment 1e).

b) The ratemap illustrations of this manipulation, Figure 5C left, show two cells, with EMD values before of 0.177 and 0.178 and after of 0.293 and 0.277. Of over 100 border cells, they show the most unrepresentative cell and the third most unrepresentative cell. Something more representative should be shown.

Following our previous comment regarding the distribution of manipulation effects, we show examples of RSC border cells that are representative of disrupted tuning due to the manipulation. This is stated explicitly in the figure legend: “Two example RSC border cells that were affected by MEC inhibition and lost their spatial tuning.”

Furthermore, as we discuss in comments 1a) and 1c), we consider the population’s mean EMD value not necessarily as ‘representative’ of the overall impact of MEC inhibition, as these methodologies did not allow for silencing all RSC-projecting neurons in MEC in awake, behaving animals.

c) Similar points as a) apply to the other two inactivation experiments using optogenetics. The cell-body inhibition results are particularly weak in effect, with laser ON EMD scores averaging 0.0192. This average is 0.001 above the relatively strict 99% threshold of 0.0191, and 0.003 above the second laser off average. Thus, the cells are on average still very border-like.

This point was partly discussed in the above comment 1a), but here, the method used for the optogenetic experiment has a particular limitation in targeting the population of cells in MEC.

First, virus expression levels and the retrograde transport efficiency of retro-AAV is not 100% (Tervo et al., 2016), and the expression is limited to a subpopulation of RSC-projecting cells in MEC (Figure 5—figure supplement 2). Next, the optic fiber of 400 µm diameter covers less than half of the lateral width of MEC, and was placed at ~0.5 mm above the dorsal edge of MEC to avoid damage to MEC cells. The laser power was 20 mW at the fiber tip, which quickly decays to 5mW/mm^2^ at 1 mm away from the fiber tip (Calculation based on the tool of the Optogenetic Resource Center; https://web.stanford.edu/group/dlab/optogenetics/). According to Choung et al., (2014), laser power of 5 mW/mm^2^ can achieve only 30% inhibition in Jaws-expressing neurons. Therefore, this method only allowed us to manipulate cells at the dorsal pole of MEC, and those in the ventral region were most likely unaffected.

Because of this methodological limitation, whether or not the mean EMD scores of the population is above or below the threshold is misleading, as some border cells would not be affected simply because projecting MEC cells were not silenced sufficiently. Our main aim for this experiment is to show that the boundary information in RSC is at least in part derived from MEC, which is reflected in changes in tuning strength (e.g., EMD scores) of RSC border cells due to inhibition of MEC. We have added an additional subpanel in Figure 5—figure supplement 3B to show these cell-specific differences between laser OFF and ON sessions.

For the revised manuscript, we further performed additional control experiments for the optogenetic manipulation, now presented in Figure 5—figure supplement 4. Two animals were implanted with an optrode in RSC, following the same procedure as for Figure 5K-O but without AAV injection. In summary, RSC border cells show high consistency in their EMD scores across sessions, unlike results shown in Figure 5O. This illustrates the significance of these optogenetic manipulations overall.

This result is now discussed in subsection “Inhibition of MEC input disrupts border coding in RSC but not vice versa” of the revised manuscript and in the legend of Figure 5—figure supplement 2.

d) as with b) Similarly unrepresentative cells it seems are shown for Figure 5L and 5N.

Please see our comment in 1b). These cells are representative of disrupted tuning due to inhibition of their inputs.

e) The disruption of firing caused by inactivation of the MEC seems slight (Figure 5D,O), and the examples in 5L (and to some extent 5N) are not convincing because the firing patterns do not seem stable across the two 'OFF' trials, so it is hard to be sure that changes in the 'ON' trial are due to the manipulation. To what extent does the laser stimulation (Figure 5O) increase the 'messyness' of firing rather than changing its tuning characteristics – eg reducing spatial information/stability or increasing excitability (are firing rates different)?

We agree with the reviewers that it is crucial to assess whether the partial disruption in the laser OFF session is due to the intrinsic instability of RSC border cells or not. We therefore performed control experiments in which laser was applied without opsin expression, and confirmed general stability of RSC border cells across the session (Figure 5—figure supplement 4). The increase of EMD scores in the laser ON, as well as partial increase in the second OFF session, is therefore specific to the MEC inactivation.

RSC border cells that are disrupted by MEC inhibition (e.g., significant increases in EMD scores in the light ON session) show only a partial recovery in the axon-terminal inhibition manipulation (Figure 5O) or a continued disruption in the cell-body manipulation in the subsequent OFF session. One possible cause for this long-lasting effect is that we used continuous laser stimulation for 5 consecutive minutes during the light ON session, and a 5 minutes break before the final laser OFF session may not be sufficient to recover normal physiological function (e.g. ionic concentration disturbance created by the chloride pumps). It is also possible that cell-body inhibition may particularly cause a long-term circuit reorganization in MEC. Contrary to changes in EMD scores, we did not observe any changes in the overall firing rates of RSC border cells due to the manipulation (Figure 5—figure supplement 3A).

We would also like to note that even stable border cells do not necessarily produce the exact same rate maps between sessions, and therefore, the direct comparison of firing fields between rate maps is misleading. Because of conjunctive sensitivity to both boundary and direction, rate maps of RSC border cells are usually blobby, which is mainly due to the animal’s directional-bias near the walls and can differ depending on the animal’s session-by-session behaviors. The EMD scores should thus give a better quantitative assessment here.

We now discuss these points in the revised manuscript in subsection “Inhibition of MEC input disrupts border coding in RSC but not vice versa”.

f) The chemogenetic and optogenetic manipulations are lacking standard controls.Specifically, there is no non-DREADDs group or sham injection recordings for the chemogenetic experiment, and there is no control virus group in the optogenetic experiments. As such the effect could be due to systemic effects in the former and with heating in the latter. The DREADDs experiments do have an internal control with the RSC-MEC reversal inactivation, but not the optogenetic experiments. That being said, the cell body inhibition experiment gives more confidence in the result.

We thank the reviewers for this important suggestion and performed additional negative control experiments in two animals against our optogenetic manipulations, now presented in Figure 5—figure supplement 4 (see also comments to 1a, 1c and previous 1e). RSC border cells did not show any significant changes in their firing rate or boundary EMD scores across sessions, as laser light was applied in the absence of an inhibitory opsin, excluding heating as a potential confound for our optogenetic manipulation results in Figure 5.

g) These findings are interpreted with exaggeration.Abstract "These egocentric representations…require inputs from MEC." Subsection “Inhibition of MEC input disrupts border coding in RSC but not vice versa” "While these DREADDs-mediated manipulation experiments suggest the necessity of MEC signals for border tuning in RSC…". Figure 5 legend: "RSC border cells require input from MEC to maintain their boundary tuning". Necessity and Require are untenable inferences from the modest effects shown, and this should all be rephrased so casual readers are not misled.They should perform a sanity-check analysis where cells with peak rates of say 1Hz are excluded from the analyses. If a cell is not really firing, it may not be that informative to examine the spatial features of the few available spikes.

We understand the reviewer’s concern and carefully rephrased these statements in the revised manuscript in the Abstract, subsection “Inhibition of MEC input disrupts border coding in RSC but not vice versa”, Discussion section and in the legend of Figure 5. While this study cannot determine whether or not the complete silencing of MEC causes a total disruption of boundary coding in MEC, our results are sufficient to conclude that MEC inputs are necessary to maintain sharp boundary tuning in RSC, as the silencing of MEC cells significantly increases EMD scores of RSC border cells, and we carefully clarified these points in the revised manuscript.

We agree that a low number of spikes can result in misleading spatial rate maps. Overall, cells are only classified as RSC border cells with an average firing rate above 0.5 Hz across all non-manipulated sessions. However, we decided to include cells that have minimal firing only in opto/chemogenetic manipulation sessions, as this is a clear indication of disrupted firing due to the manipulation.

h) Subicular boundary related inputs. Boundary coding being both preserved in darkness (Lever et al., 2009; see also Brotons-Mas et al., 2010), and most cells maintaining their tuning without walls present (Lever et al., 2009; Stewart et al., 2014) is shown in the subiculum and thus there is a source of boundary-coding information additional to the Entorhinal cortex that shares some key features with these retrosplenial border cells. The projection to the retrosplenial cortex from the dorsal subiculum, where boundary vector cells have been found, is dense (see e.g. Wyss and Van Groen, 1992). I think Rosene and van Hoesen, 1977 suggest the main cortical afferent to the granular RSC originates in the subiculum. Thus, consideration of boundary information coming into the RSC should mention such boundary cell and anatomy tracing work.

We agree with the reviewers that the subiculum is indeed a promising candidate to provide boundary-related inputs for RSC border cells, besides MEC, in particular because of its dense connectivity and the presence of boundary vector cells. It is possible that some of the unperturbed RSC border cells after the pharmaco- or optogenetic silencing of MEC may receive inputs from the subiculum.

We note that RSC border cells fire predominantly at the proximity of walls (Figure 4—figure supplement 1C,D) unlike the vector-like representation of boundary vector cells in the subiculum, which indicates higher similarity to MEC border cells. However, some key features of boundary vector cells, in terms of the maintenance of boundary tuning in darkness or without walls present, are shared with RSC border cells. Further investigation into the specific contribution of subicular inputs to RSC is important future work, and we have added a section in the discussion section to consider the potential relevance of subicular inputs, including the references provided by the reviewer.

2) What are the defining characteristics of the RSC 'border cells', are they directionally tuned, how do they relate to other boundary-responsive cells, and what to call them?a) Quantification of border scores is by comparison to a Gaussian smoothed template of firing at the borders. However, the comparison method (earth mover distance, EMD) is not clear – giving an intuitive explanation, such as the total distance moved by all units of firing rate to match the firing rate and template distributions would be helpful. More intuition for the numbers would be gained by showing cells with values near the classification thresholds, not just at/near the tails. Figure 1F shows values of 0.14, 0.145, 0.159…. and then 0.222 and 0.312. Please show cells near 0.1906 cutoff. Relatedly, in Figure 1E, show the EMD values corresponding to 95 and 90% cutoffs.

We agree with the reviewers that a more intuitive explanation on the EMD metric would benefit the reader’s understanding, in particular because it is a novel methodology. We have added an additional explanation in the Results section. We further added many example rate maps of cells with EMD scores at evenly spaced intervals in the range of 0.14 to 0.23 in Figure 1—figure supplement 2G from a single animal’s dataset, and provided the 95^th^ and 90^th^ percentile values in Figure 1E to give a better intuition on the relationship between a cell’s ratemap and its associated EMD score.

b) It is not clear the extent to which spiking has to be restricted to the borders of the environment, how the method captures spiking that is displaced a certain distance from a border, and how the distance and egocentric direction tuning of each cell was found. If the template is only at the border are more distally tuned cells missed?Is this same measure applied to the MEC (for fair comparison of the MEC and RSC it should be)? And does it find cells that fire distant from the border in MEC? This is particularly relevant given the puzzle that spatial representation occurs up to 50cm from the border by MEC 'border cells'? Did distance tuning differ between RSC and MEC (please show the distance tuning distributions for both areas)?

The EMD method is somewhat flexible in cell classification by allowing neurons to have a certain degree of spikes in the center of the arena. It focusses on the general weight of the firing fields and assigns increasingly higher distance value when this weight moves away from the edges. This can be seen in the newly added rate maps in Figure 1—figure supplement 2G (see also previous comment 2a), where rate maps have increasingly more firing fields towards the center as their EMD score increases, and cells generally have a large proportion of spikes away from the walls when the EMD falls above the 1^st^-percentile threshold of 0.191.

To assess the procedure’s ability to classify cells with different preferred firing distances, we simulated a set of synthetic rate maps to quantify this relationship between the wall distance of spikes and a cell’s EMD score (Figure 4—figure supplement 1C). As expected, the EMD score is minimal when all spikes are nearby the wall, with non-zero values and offset of the trough due to behavioral under-sampling of space. We further observed a linear increase in EMD scores as a function of increasing distance of the spikes away from the wall, and simulated cells would be classified until spikes reached the threshold of 17-18 cm wall distance.

Therefore, as the reviewers pointed out, it is possible that the original boundary template might have missed border cells with a larger distance tuning to the walls, because this template was optimal for capturing cells with preferred distance tuning below 20 cm. In order to assess the presence of border cells that fire beyond this range, we used 5 additional templates with their main field at increasing distance away from the wall (Figure 4—figure supplement 1D; this was done only for a subset of data due to computational constraints).

The result suggests that cells that were identified with the original template could still be captured with templates that have fields at 2 rows distance (e.g., up to template 3). Templates at these distances would capture a subset of additional cells, which were qualitatively similar to the original border cells (firing fields attached to the wall but more extended, with a peak distance tuning of 16 ± 5 cm). Importantly, as the number of cells identified with the original template decreased toward zero, we did not see any new cells identified with templates at far distances away from the wall either. This result confirms that RSC border cells were mostly captured by the original template, exhibiting distance tuning at the proximity of walls up to 20 cm, and the existence of RSC cells with longer wall distance tuning is unlikely.

It is true however that we could employ the same EMD procedure to categorize MEC border cells, but applying this procedure is not straight forward. MEC cells are typically tuned to only one or two walls, and it would thus require EMD scores for multiple templates that have firing fields at different combinations of them (10 templates in total, or 14 if you consider 3-wall combinations). For the statistical tests this would require the comparison between a shuffled distribution of each respective template and the cell’s EMD score, which after adjusting for multiple comparisons would yield rather conservative estimates.

We have performed a preliminary exploration of this approach using templates with firing fields attached to only 1 wall (Author response image 1). What we observed is that the EMD procedure was able to find cells with rate maps that are very similar to the template, some of which were identified previously as border cells (Author response image 1, middle and bottom rows). However, this approach also identified a number of false positives, including cells that have only small firing fields nearby the wall, and the two distributions of EMD scores between border and non-border cells are partially overlapping (Author response image 1). These examples, together with the complications of elaborate statistical comparisons, demonstrate that the procedure would need substantial adaptations to be suitable for classification of MEC border cells. Unfortunately, this undermines the benefit we’re trying to achieve here, which is greater comparability with our results in RSC. We thus have a strong preference to continue to rely on the original border score for border cell classification in MEC.

**Author response image 1. sa2fig1:** EMD classification of border cells in MEC using MEC-specific templates. (A) Distribution of EMD scores, selecting the lowest value on any of the four templates for each MEC neuron. Dark green represents the population of border cells in MEC that were classified previously using the border score. (B) The four templates used in this procedure, with firing fields attached to a single wall. (C) Spatial rate maps of 8 cells with the lowest EMD score on any of the templates (e.g., most left values in A). Top row: two examples of cells with localized firing nearby the wall, which resulted in a low EMD score. Middle row: three examples of cells not classified by the border score (e.g. values below 0.5), but still rather border-like. Bottom-row: three border cells identified by both methods.

As to why our decoder performed better at higher wall distances in MEC cells compared to RSC (Figure 6C), we believe it’s because of a different overall spike distribution towards the center of the arena. Despite both populations of border cells having maximal firing at the edge of the arena (see peak distance tuning distributions in Figure 4J for RSC and Figure 6—figure supplement 1E for MEC), we observed significantly lower population vector correlations in MEC at a longer wall distance range (Figure 6—figure supplement 1B). What this suggests is more variability between cells, where some show stronger decays of firing rate as the animal moves away from the wall, while others have weaker decay. Furthermore, decoding performance is dependent on reproducibility of the same firing rate at the same distance, which is not necessarily apparent in a rate map, but which produced more consistent results for MEC border cells.

Regarding our procedure for obtaining peak tuning values, this was previously described in the methods > border rate maps section: “A cell's preferred direction and distance was obtained by finding the bin with maximal firing rate, and selecting the bin's corresponding distance and angle values.”

These points are now described in the Results section of the revised manuscript.

c) The comparison to actual border cells (that must fire continuously along a border) is important – the new score does not penalise gaps in firing (hence the appearance of a grid cell in Figure 5L top left?), nor does it require an allocentric tuning direction (a characteristic of border cells and boundary vector cells).How strong is the tuning to egocentric direction or are these cells that mostly fire near a border in any direction? 185/485 egocentrically tuned cells seems low. Do 300/485 have no directional modulation, or is there qualitative egocentric modulation but below the statistical threshold? If not directionally modulated at all they can't be classified as either egocentric or allocentric.The claims made (see also 2d) warrant further investigation of the potential differences between their confirmed egocentric border cells and the potentially numerous allocentric border cells within the RSC. Please provide the distribution of egocentric (and allocentric) directional tuning strengths across the populations of 'border cells' in RSC and MEC.

We would like to clarify that RSC border cells described in this work do fire alongside the entire wall, albeit not in equal proportion. Many cells are constrained by additional factors, like direction of the wall relative to the animal which we explore in Figure 4, that causes firing to appear discontinuous in a cell’s rate map. One potential cause is a substantial uneven distribution of both position and direction occupancies across spatial bins, as animals naturally sample the environment unevenly in a short recording session. Illustrations of this phenomenon are the simulated border cells presented in Figure 6—figure supplement 1F, as well as the higher EMD scores in the 0-5 cm distance range in Figure 4—figure supplement 1C. Spikes were generated to represent a pure border cell, with the additional constraint of a boundary within a certain range of directions (width is π/2) relative to the animal. The resulting trajectory-spike plots and rate maps are discontinuous, and the top cell in this figure has no spikes in the north-west corner due to under-sampling. Generally speaking, most trajectory-spike plots show spikes covering the entire wall however (see examples in Figure 1C, Figure 4A and 4E-F, and Figure 4—figure supplement 2), and for directionally-tuned neurons the egocentric border maps have a single continuous firing field (Figure 4C and 4E-F, and Figure 4—figure supplement 2).

Regarding the tuning strength of all border cells, we indeed found that roughly 40% of all neurons have egocentric directional tuning. We now provide the proportions of egocentric and allocentric directionally-tuned border cells in RSC and MEC in Figure 6B. This result indicates that only 7% of all RSC border cells are tuned to allocentric head-direction, which is similar to non-border cells (7.3%, also see minor point 1). Therefore, RSC border cells below the egocentric statistical threshold are not necessarily allocentric, and still have relatively high MVL for egocentric tuning (RSC: directional cells, MVL = 0.40 ± 0.01; non-directional cells, MVL = 0.29 ± 0.01; mean ± S.E.M.; Wilcoxon ranksum test: z = 9.45, p = 3.3 x 10^-21^). In contrast, a subset of border cells in MEC show conjunctive coding with allocentric head-direction, as previously reported in Solstad et al. (2008), where 58% of cells are significantly tuned to allocentric HD.

These results highlight the difference between egocentric-dominant cells in RSC and allocentric-dominant cells in MEC, and we included this point in subsection “RSC border coding is more local and correlated with the animal’s future motion”.

d) Clarification in the language used in the Abstract, Introduction, and Discussion section seems vital. The authors make much of the distinction between the allocentric boundary cells in other regions, and the egocentric boundary cells here. Furthermore, the abstract offers the summary: 'Border cells in RSC…are sensitive to the animal's direction to nearby borders'. Is Earth Mover Distance (EMD) alone egocentric? If not, and all of the analyses are on the EMD population, the result should not be framed as allocentric to egocentric transformation. If the egocentric border cells were analyzed throughout, that would justify the title and framing.It is confusing to refer to both the barrier-responsive cells in RSC and the previously documented EC border cells as simply 'border cells', when the two populations appear to be different. 'Border cells' were defined by Solstad et al., 2008 as cells that fire when the animal is right next to a barrier in a specific allocentric direction (thus distinguishing them from the pre-existing 'boundary vector cells'). They subsequently suggested that border cells respond to direct contact with a physically present barrier, unlike 'object vector cells' which also respond to an object suspended above them (Hoydal et al., 2019), or boundary vector cells which can respond to the previous location of a barrier (Poulter et al., bioRxiv). The barrier-responsive RSC cells are clearly not 'border cells' – they can respond to barriers at a distance, some with a tuning to egocentric direction, and (the authors argue) are not a direct sensory response to the barrier.What should they be called? If they think the population is generally tuned to egocentric direction (this is not clear, see next point below) 'egocentric boundary vector cells' would be technically correct, but is a bit of a mouthful, the main thing to avoid would be calling them something they are not. Perhaps referring to them by a label containing 'RSC' would at least make clear when they are referring to their RSC cells and when they are referring to border cells in EC ('RSC boundary cells' or similar).

We completely agree with the reviewer that the nomenclature of functional cell types that encode aspects of environmental boundaries is becoming complicated. The first use of ‘border cells’ was proposed by Solstad et al., (2008), and refers to a group of cells in MEC with specific boundary tuning properties. The definition itself however is not very informative of their exact properties, other than the encoding of border information. Conversely, the ‘boundary vector cell’ naming implies vector-like properties (e.g. distance and direction) and refers to boundary-responsive cells found in the Subiculum. As the reviewer suggests, the term ‘egocentric boundary vector cells’ is another potential definition for the direction-selective cells, and this has been adopted by Alexander et al., (2020) together with ‘inverse egocentric boundary vector cells’ for boundary-off cells.

It’s important to reiterate however that only 39% of boundary-responsive cells reported in this work show directional wall tuning, making the ‘egocentric’ term rather inaccurate, as opposed to 100% of the cells reported in Alexander et al., (2020). Instead, all cells in this manuscript are identified by our EMD procedure that share one defining feature, that is, a strong degree of spiking near all outer walls of the arena and are thus boundary-responsive across the environment. Most of our analyses, with the exception of Figure 4, have focused on this distance aspect of cell tuning, and border cells in both RSC and MEC share wall-proximity tuning properties (Figure 4J, Figure 6—figure supplement 1E, see our comments to 2a). Our additional analyses with different EMD templates further confirmed that the existence of RSC border cells at a longer wall-distance tuning is unlikely.

We thus opted for a more general term of ‘border cells’, and added the anatomical label of RSC or MEC throughout this work.

In light of this discussion we have reviewed the terminology throughout the manuscript and adapted text wherever necessary to reduce the amount of potential confusion between different functional cell types and/or anatomical regions. We would be happy to discuss further if the reviewers still consider the terminology confusing in the revised manuscript.

e) Description of MEC border cells is a little misleading. "These features are shared with MEC border cells, as their boundary tuning is also maintained without walls present (Solstad et al., 2008)".The data of Solstad et al., 2008 (Figure S8 and S9) show rather altered firing when walls are removed. (A) Of 10 border cells studied in the wall-no wall manipulation, only one maintained its field in the no wall environment, the others did complex or seemingly rotational remapping. (B) Furthermore, Solstad S9 suggests that the seeming-rotational remapping of border fields occurred despite the directional firing of simultaneously recorded head direction cells' being stable throughout the wall-no wall manipulation. Thus, even the seemingly simple rotational remapping may be more complex. In all, this section thus needs revision and clarification.

The main result of the no-wall manipulation in Solstad et al., (2008) is described in Figure S7. We have followed the original authors’ interpretation of their data: “Border fields were often but not always maintained after removal of the external walls”.

The reviewer rightfully points out however that border cells in MEC show rotational remapping to some degree after wall removal, which is a complex dynamic and its interpretation depends on a thorough understanding of the underlying circuitry involved (e.g. relationship between border cells in MEC and head-direction, grid and place cells). Yet rotational remapping does not alter inherent firing properties of border cells as they still encode distance information to nearby boundaries, although to a different border. In that sense the features described in our results are in line with those obtained in Solstad et al., (2008).

We have revised the Discussion section to clarify these subtle differences between both results.

3) 'Complete Darkness'. If a darkness condition disrupts, there is less burden on the experimenter to ensure its completeness; e.g. darkness greatly reducing gridness implies vision aids grid cell firing in mice (Chen et al., 2019) is a safe inference, and if the darkness was not complete, this does not matter that much. […] For evidence that humans, rats, mice and cats can see in the supposedly invisible infrared spectrum, if they are dark-adapted, see e.g. Pardue et al., 2001 and Palczewskaet al., 2014. A paper on good ferret vision at 870nm may be of interest (Newbold and King, 2009).

We understand the reviewer’s concerns regarding our definition of complete darkness. These experiments were performed under the assumption that rats have limited vision in the higher wavelengths, with cone sensitivity tapering off rapidly above 600-650 nm (e.g. in the range of red light; see spectral sensitivity functions in Figure 1, Figure 2, Figure 3 in the review of Jacobs et al., 2001). No video-based tracking system can work in the absence of any light, so instead of using the visible light spectrum, our recording set-up included 6 Flex3 cameras with many infra-red (IR) LEDs aimed at the arena, tracking IR-reflective markers connected to the implanted electrode.

We have taken several measures during the experiment to ensure no visible light was present for the animals. This includes taping off small light sources in the room, such as computer and sensor lights, while the arena was fully enclosed by a thick, black curtain. A room lamp was turned on for dimly light conditions until 10 seconds before the start of the recording, and turned on again during the inter-trial interval duration of ~5 minutes. During recording, the experimenter’s experience was that of complete darkness, with zero visibility in the room even after 15 minutes of adaptation. He remained stationary and silent near the arena throughout the recording while scattering food pellets.

However, it comes as a surprise to us that there is evidence that both rats and humans have a sensitivity to some degree in the infra-red spectral range after dark-adaptation. In order to compare our IR light conditions with those references provided by the reviewer, we have taken additional measurements to capture the spectral density of our lighting conditions, together with an energy intensity measure at the peak wavelength (Author response image 2).

**Author response image 2. sa2fig2:** Spectral density measures of the light used in the experiments, captured by a Flame-S spectrometer (OceanOptics). The black line represents the spectral component of the infra-red LED used for tracking during darkness, while the red line shows components of the desk lamp used to create dim light conditions. Intensity unit sizes are arbitrary, with the sensor kept at a distance of the light source as to not saturate it.

The IR LED illumination peaks at 850 nm, and tapers off rapidly with little to no energy remaining below the range of 750 nm. While there is no overlap between this spectral range and the rat’s eye spectral sensitivity functions in the traditional literature (Jacobs et al., 2001), this range does cover the infra-red stimulation used in the literature cited by the reviewer.

In particular, Pardue et al., (2001) report visual-evoked responses in V1 of rats as a result of direct retinal stimulation during anaesthesia using LEDs at peak wavelengths of 890 and 936 nm. Palczewska et al., (2014) investigate human detection of direct retinal stimulation using fs-pulsed laser light in the range of 950-1200 nm, while Newbold and King (2009) study ferret vision where animals learned to detect IR LEDs turned on in the range of 870 nm.

We would like to highlight two important differences however between these studies and ours:

1) Light intensity. All three studies used high light intensities, directly stimulating the retina at close distances and after dark-adaptation. We have measured light intensity of our LED system using a S170C photodiode (Thorlabs) and PM100D power meter (Thorlabs), which came in at 6000 µW or 18.52 W/m^2^ when sampled near the light source. Our cameras were positioned 2 m above the arena surface from a ceiling mount, at a 45-60° angle pointing downwards, and the energy at the arena surface was substantially lower, measuring 0.51 ± 0.06 W/m^2^. Given the presence of 6 cameras, this is a 18.52*6/0.51 = 217-fold reduction in light energy entering the eye of the animal due to light divergence over distance.

2) Light detection. There is a real difference between the ability to detect the presence or absence of a point light source, as opposed to using that light to see the environment. In the first case, detection can occur in extremely low visibility conditions, while the latter requires much higher levels of illumination.

Finally, as a sanity check, we further checked whether cells in RSC changed their spiking rates as a function of dark-adaptation, but we could observe no significant differences in the number of spikes between the first half and second half of each darkness session. This holds true both for border cells (Wilcoxon signed rank test: z = -1.42, p = 0.156) and other cells (Wilcoxon signed rank test: z = 1.04, p = 0.300) in RSC.

Taken together we hope this provides enough support for our interpretation that our animals were in no position to have enough visibility under these IR-light conditions to properly see their surroundings. All these experimental conditions are explicitly mentioned in subsection “Border cells retain th 214 eir tuning in darkness and are not driven directly by whisker sensation”, and we have added further descriptions in the Materials and methods section of the revised manuscript.

4) Object insertionThis manipulation is not yet convincing.a) Though it features in their abstract as a main finding, there are not many cells for this experiment.b) It seems sub-optimal that the ROI around the object is square, not circular, and quite a large square.

The reviewer highlights an issue here that we ourselves have struggled with too. That is, the observation that RSC border cells did not increase their firing in an area around the new object (Figure 2G-I), while they did show an increase in the boundary template EMD scores (Figure 2J), suggestive of changes in the cell’s rate map after introducing an object.

However, please note that our EMD method is sensitive enough to detect changes elicited by an added object even in a small number of cells, illustrated by the analysis of object manipulations in S1bf neurons, where EMD scores of 23 cells dropped significantly as a result of object insertion (Figure 2—figure supplement 1H). Furthermore, a change in firing due to the object in either direction (increased or decreased) would show in our measure of spatial correlations between session types (Figure 2I;), but correlations did not decrease for RSC border cells, in contrast to S1bf neurons (Figure 2—figure supplement 1I, left panel). Here, the total number of cells was equal in both cases (23 classified cells) and we used the same approach and statistical tests, yet found a significant shift of the firing fields by the object insertion only in the somatosensory cortex, but not RSC.

Regarding the ROI drawn around the object location, we realized the illustration in Figure 2F was not an accurate representation of the underlying region used for the calculations, as the ROI had a width of 8/25 bins (e.g. 32 cm, or 1/3^rd^ of the arena) but was drawn larger in the figure (more than half width). In addition, we took particular care as to not include the firing fields close to the nearby walls in the north-west corner. Considering the reviewer’s comment, we have adapted this ROI to now be of circular shape, with a diameter 8 spatial bins, covering 11 cm of space on each side of the object. We have updated Figures 2F and 2H accordingly, but the results, described in subsection “Border cells form new firing fields nearby added walls but not objects”, have not changed qualitatively.

c) Perhaps 10 cells increase their firing in this large square ROI, and others decrease their firing. A lack of perturbation by object insertion is not self-evident from these data; rather there could be some cell-specificity, with some cells with firing being actively inhibited by the object, and others being excited by it. There seem to be quite large reductions in firing in 3 cells.d) Thus, their aggregate analysis is perhaps a little simplistic, and misses out cell-specific responses. As there is not much statistical power, it might be simpler just to show the majority of these cell responses in a supplementary figure.

Different neurons indeed show variability in firing inside the ROI between session types. While border cells generally fire near the outer walls, a substantial number of spikes can still be observed in the center area even in an open field, which are considered ‘noisy’ spikes. This activity has high variance both between sessions, and between cells, which can be seen when computing the same ROI result between the first and last regular session without on object in the arena (Normalized variance in FR inside ROI between sessions: Reg-Reg, CV = 0.93; Reg-obj, CV = 1.06; Figure 2H, Author response image 3).

What we observed in Figure 2H is that across the population, RSC border cells do not fire consistent with their firing patterns nearby walls (e.g. formation of new firing fields, as seen in Figure 2B,C with walls). Furthermore, a combination of cell-specific increases and decreases between session types would result in decreased spatial correlations, which we did not observe in Figure 2I. Taken together we think such cell-specific changes are unlikely and we do not analyze further in this direction.

**Author response image 3. sa2fig3:** Firing rate of border cells inside the object ROI between the first and last regular session without an object present. Individual cells (grey lines) show both increased and decreased firing, but no mean difference was found between the sessions, as expected (First regular, FR = 1.61 ± 0.23; Last Regular, FR = 1.50 ± 0.29; Wilcoxon signed rank test: z = -0.763, p = 0.45).

e) Importantly, the EMD templates are biased towards increasing the likelihood of a null finding. Put simply, the walls in the boundary template have two rows of high firing bins near them, whereas in the object template, these same walls have only one row of high firing bins near them, and moreover this row is of lower rate. In marked contrast, the object in the object template has three rows of higher rate bins around it. (To be clear, the authors mention this bias in their Materials and methods section: "adding additional weight in the location of placed objects/walls".) Thus, the object is expected (by the template) to elicit high firing for a more extended distance and at higher rates than that at the boundary even though the boundary is an extended cue and should influence the cell more. There is no need for such a biased hypothesis.

We understand the reviewers’ concern of the weight distributions of EMD templates, but would like to underscore the notion that the exact distribution of weight in such a template is not a reflection of our underlying hypothesis of the rate distributions of classified cells. The main aim of this object template is to determine whether or not neurons form firing fields around the object location in a similar fashion as they do to walls. To that extent, adding additional weight near the object would increase, rather than decrease, the likelihood of a significant finding if the object elicits additional spikes from the neuron when comparing the object template scores between regular and object sessions. However, we still did not find a significant decrease in EMD scores with this template for RSC border cells, contrary to the S1bf neurons, consistent with the ROI spiking rate and spatial correlation results.

We have repeated the EMD analysis with another template with equal distribution of weight around the object and walls (1 row at equal values to the outer walls), and the resulting EMD distribution is qualitatively the same as the template for Figure 2J, with no significant differences between regular and object sessions (Normalized object EMD score: R1, 1.0 ± 0, O1, 1.036 ± 0.032, O2, 0.986 ± 0.034, R2, 1.009 ± 0.028; Friedman test: Χ2(3) = 4.25, p = 0.24).

f) More details should be provided as to the previous experience with the objects. Might there be an inhibition by novelty? g) Object size: Figure 2F says the object was 15cm in diameter, but Figure 2—figure supplement 1, part g and the main text says 10cm. Please check and clarify sizes for all experiments, including the size of the ROI around the object.

The correct object size was 10 cm and we have corrected the experimental details in the method section, and reported ROI sizes for the added wall and object sessions in the revised manuscript in subsection “Border cells form new firing fields nearby added walls but not objects”.

In summary, there is no doubt whatsoever that the walls exert a greater influence than the object, but that is different from saying that "firing of RSC border cells…is invariant to an object introduced into the maze" (Discussion section). This is not an accurate summary when even their analysis as it stands shows a substantial increase in EMD to the boundary template in the object condition.

Our interpretation of the data is that RSC border cells do not exhibit a significant shift of their firing fields towards the proximity of an introduced object location, not like an introduced wall, supporting the different coding mechanism between objects and walls. However, as the reviewers pointed out, it does raise the question of what were the exact changes in the rate maps that resulted in increased boundary template EMD scores. These changes appear not to be specific to the object location, and one possible explanation could be a behavioral bias for the animal after introducing an object to the arena, as we have shown a modulation on FR by turning behavior in RSC border cells (Figure 6E-I). However, substantially more data and in-depth experiments would be required to clarify the interactions between objects and walls, which is something we leave for a future research project.

Following the reviewers’ comment that our original statements such as “invariant to an object” were too strong of nature and do not reflect our current results, we have updated several sections of the main text to better accommodate their concerns and our views listed above, and rephrased our conclusion in subsection “RSC border coding is more local and correlated with the animal’s future motion”.

[Editors' note: further revisions were suggested prior to acceptance, as described below.]

The manuscript has been improved but there are two remaining issues that need to be addressed before acceptance, as outlined below:1) The statement in the Abstract where it says the cells "depend on inputs from MEC" should be moderated to something like "are influenced by inputs from MEC" to be more consistent with the point made in the reviews.2) In the response you say, "we decided to include cells that have minimal firing only in opto/chemogenetic manipulation sessions, as this is a clear indication of disrupted firing due to the manipulation." Please confirm the process by which the population of cells was selected on which to test the effect of the manipulation.

Regarding the first point about the Abstract, we now changed the statement from (the cells) “depends on” to “are affected by” inputs from MEC”. (We also made small changes in other part of the Abstract due to the 150 word limit).

On the second point concerning about the firing rate threshold, we now clarified the description and further performed additional analysis. As described in the point 1g) of the rebuttal letter, we used a firing rate threshold of 0.5 Hz only for the baseline sessions in the original analysis. In this revised manuscript, in order to assess whether the main effect of increased EMD scores could be explained by low-firing cells, we repeated the analyses with the consistent 0.5 Hz threshold throughout all sessions and obtained the same conclusions. This additional anaysis confirms that the impairment of boundary coding cannot simply be explaine by the reduction of spiking. This point is now described in subsection “Inhibition of MEC input disrupts border coding in RSC but not vice versa” and subsection “Spike sorting and cell classification” of the revised manuscript.